# FEDERATED LEARNING FROM ONLY UNLABELED DATA WITH CLASS-CONDITIONAL-SHARING CLIENTS

**Nan Lu**[1]  **Zhao Wang**[2]  **Xiaoxiao Li**[3]  **Gang Niu**[4*]  **Qi Dou**[2]  **Masashi Sugiyama**[4,1]
[1]The University of Tokyo   [2]The Chinese University of Hong Kong
[3]The University of British Columbia   [4]RIKEN
{lu@edu.,sugi@}k.u-tokyo.ac.jp, {zwang21@cse.,qidou@}cuhk.edu.hk
xiaoxiao.li@ece.ubc.ca, gang.niu.ml@gmail.com

## ABSTRACT

Supervised *federated learning* (FL) enables multiple clients to share the trained model without sharing their labeled data. However, potential clients might even be *reluctant* to label their own data, which could limit the applicability of FL in practice. In this paper, we show the possibility of *unsupervised* FL whose model is still a classifier for predicting class labels, if the class-prior probabilities are *shifted* while the class-conditional distributions are *shared* among the *unlabeled* data owned by the clients. We propose *federation of unsupervised learning* (FedUL), where the unlabeled data are transformed into *surrogate* labeled data for each of the clients, a *modified* model is trained by supervised FL, and the *wanted* model is recovered from the modified model. FedUL is a very general solution to unsupervised FL: it is compatible with many supervised FL methods, and the recovery of the wanted model can be theoretically guaranteed as if the data have been labeled. Experiments on benchmark and real-world datasets demonstrate the effectiveness of FedUL. Code is available at https://github.com/lunanbit/FedUL.

## 1 INTRODUCTION

*Federated learning* (FL) has received significant attention from both academic and industrial perspectives in that it can bring together separate data sources and allow multiple clients to train a central model in a collaborative but private manner (McMahan et al., 2017; Kairouz et al., 2019; Yang et al., 2019). So far, the majority of FL researches focused on the *supervised* setting, requiring collected data at each client to be *fully labeled*. In practice, this may hinder the applicability of FL since manually labeling large-scale training data can be extremely costly, and sometimes may not even be possible for privacy concerns in for example medical diagnosis (Ng et al., 2021).

To promote the applicability of FL, we are interested in a challenging *unsupervised FL* setting where *only unlabeled* (U) data are available at the clients under the condition described blow. Our goal is still to train a classification model that can predict accurate class labels. It is often observed that the clients collect their own data with different temporal and/or spatial patterns, e.g., a hospital (data center) may store patient data every month or a particular user (mobile device) may take photos at different places. Therefore, we consider a realistic setting that the U data at a client come in the form of *separate data sets*, and the data distribution of each U set and the number of U sets available at each client may *vary*. Without *any* labels, it is unclear whether FL can be performed effectively.

In this paper, we show the possibility: the aforementioned unsupervised FL problem can be solved if the class-prior probabilities are shifted while the class-conditional distributions are shared between the available U sets at the clients, and these class priors are known to the clients. Such a learning scenario is conceivable in many real-world problems. For example, the hospital may not release the diagnostic labels of patients due to privacy concerns, but the morbidity rates of diseases (corresponding to the class priors) may change over time and can be accessible from public medical reports (Croft et al., 2018); moreover, in many cases it is possible to estimate the class priors much more cheaply than to collect ground-truth labels (Quadrianto et al., 2009a; Sugiyama et al., 2022).

---

*Correspondence to: Gang Niu <gang.niu.ml@gmail.com>.

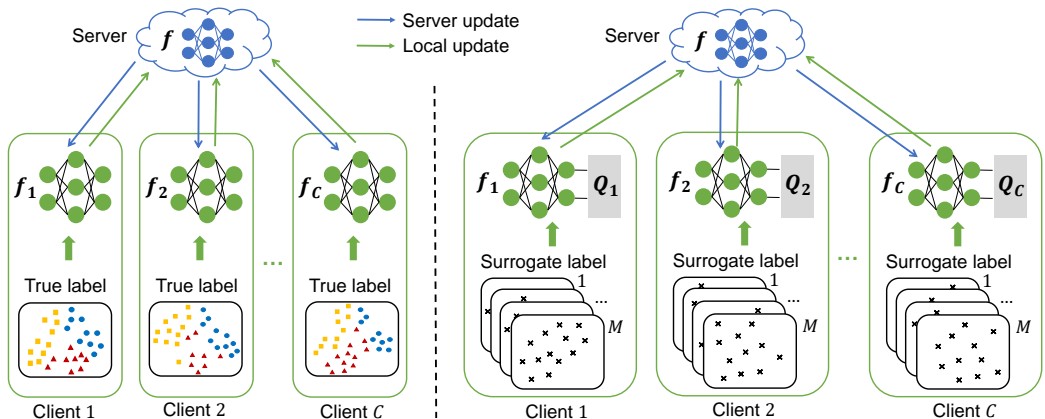

In the left panel, each client $\boldsymbol{f}_c$ ($c \in [C]$) has access to *fully labeled data*, and a global model $\boldsymbol{f}$ can be trained using a supervised FL scheme, e.g., Federated Averaging (FedAvg) (McMahan et al., 2017). In the right panel, each client has access to only unlabeled (U) data coming in the form of separate sets, where supervised FL methods cannot be directly applied. We propose FedUL that treats the indexes of the U sets as *surrogate labels*, transforms the local U datasets to (surrogate) labeled datasets, and formulates a surrogate supervised FL task. Then we modify each client model to be compatible with the surrogate task by adding a fixed transition layer $\boldsymbol{Q}_c$ to the original model output $\boldsymbol{f}_c$. FedUL can incorporate existing supervised FL methods as its base method for surrogate training, and the wanted model $\boldsymbol{f}$ can be retrieved from the surrogate model.

Figure 1: Illustration of standard supervised FL scheme vs. proposed FedUL scheme.

Based on this finding, we propose the *federation of unsupervised learning* (FedUL) (see Figure 1). In FedUL, the original clients with U data are transformed to surrogate clients with (surrogate) labeled data and then supervised FL can be applied, which we call the surrogate task. Here, the difficulty is how to infer the wanted model for the original classification task from the model learned by the surrogate task. We solve this problem by bridging the original and surrogate class-posterior probabilities with certain injective transition functions. This can be implemented by adding a specifically designed transition layer to the output of each client model, so that the learned model is guaranteed to be a good approximation of the original class-posterior probability.

FedUL has many key advantages. On the one hand, FedUL is a very general and flexible solution: it works as a wrapper that transforms the original clients to the surrogate ones and therefore is compatible with many supervised FL methods, e.g., FedAvg. On the other hand, FedUL is computationally efficient and easy-to-implement: since the added transformation layer is fixed and determined by the class priors, FedUL adds no more burden on hyper-parameter tuning and/or optimization. Our contributions are summarized as follows:

- Methodologically, we propose a novel FedUL method that solves a certain unsupervised FL problem, expanding the applicability of FL.
- Theoretically, we prove the optimal global model learned by FedUL from only U data converges to the optimal global model learned by supervised FL from labeled data under mild conditions.
- Empirically, we demonstrate the effectiveness of FedUL on benchmark and real-world datasets.

**Related Work:** FL has been extensively studied in the supervised setting. One of the most popular supervised FL paradigms is FedAvg (McMahan et al., 2017) which aggregates the local updates at the server and transmits the averaged model back to local clients. In contrast, FL in the unsupervised setting is less explored. Recent studies have shown the possibility of federated clustering with U data (Ghosh et al., 2020; Dennis et al., 2021). However, these clustering based methods rely on geometric or information-theoretic assumptions to build their learning objectives (Chapelle et al., 2006), and are suboptimal for classification goals. Our work is intrinsically different from them in the sense that we show the possibility of federated classification with U data based on *empirical risk minimization* (ERM) (Vapnik, 1998), and therefore the optimal model recovery can be guaranteed.

Learning from multiple U sets has also been studied in classical centralized learning. Mainstream methods include *learning with label proportions* (LLP) (Quadrianto et al., 2009b) and $U^m$ *classification* (Lu et al., 2021). The former learns a multiclass classifier from multiple U sets based on

*empirical proportion risk minimization* (EPRM) (Yu et al., 2014), while the latter learns a binary classifier from multiple U sets based on ERM. EPRM is inferior to ERM since its learning is not consistent. Yet, it is still unexplored how to learn a multiclass classification model from multiple U sets based on ERM. To the best of our knowledge, this work is the first attempt to tackle this problem in the FL setup, which is built upon the binary $U^m$ classification in centralized learning. A full discussion of related work can be found at Appendix B.

## 2  STANDARD FEDERATED LEARNING

We begin by reviewing the standard FL. Let us consider a $K$-class classification problem with the feature space $\mathcal{X} \subset \mathbb{R}^d$ and the label space $\mathcal{Y} = [K]$, where $d$ is the input dimension and $[K] := \{1, \ldots, K\}$. Let $\boldsymbol{x} \in \mathcal{X}$ and $y \in \mathcal{Y}$ be the input and output random variables following an underlying joint distribution with density $p(\boldsymbol{x}, y)$, which can be identified via the class priors $\{\pi^k = p(y = k)\}_{k=1}^K$ and the class-conditional densities $\{p(\boldsymbol{x} \mid y = k)\}_{k=1}^K$. Let $\boldsymbol{f} : \mathcal{X} \to \mathbb{R}^K$ be a classification model that assigns a score to each of the $K$ classes for a given input $\boldsymbol{x}$ and then outputs the predicted label by $y_{\text{pred}} = \operatorname{argmax}_{k \in [K]}(\boldsymbol{f}(\boldsymbol{x}))_k$, where $(\boldsymbol{f}(\boldsymbol{x}))_k$ is the $k$-th element of $\boldsymbol{f}(\boldsymbol{x})$.

In the standard FL setup with $C$ clients, for $c \in [C]$, the $c$-th client has access to a labeled training set $\mathcal{D}_c = \{(\boldsymbol{x}_i^c, y_i^c)\}_{i=1}^{n_c} \sim p_c(\boldsymbol{x}, y)$ of sample size $n_c$ and learns its local model $\boldsymbol{f}_c$ by minimizing the risk

$$R_c(\boldsymbol{f}_c) := \mathbb{E}_{(\boldsymbol{x}, y) \sim p_c(\boldsymbol{x}, y)}[\ell(\boldsymbol{f}_c(\boldsymbol{x}), y)]. \tag{1}$$

Here, $\mathbb{E}$ denotes the expectation, $\ell : \mathbb{R}^K \times \mathcal{Y} \to \mathbb{R}_+$ is a proper loss function, e.g., the *softmax cross-entropy loss* $\ell_{\text{ce}}(\boldsymbol{f}_c(\boldsymbol{x}), y) = -\sum_{k=1}^K \mathbf{1}(y = k) \log \left( \frac{\exp((\boldsymbol{f}_c(\boldsymbol{x}))_k)}{\sum_{i=1}^K \exp((\boldsymbol{f}_c(\boldsymbol{x}))_i)} \right) = \log \left( \sum_{i=1}^K \exp((\boldsymbol{f}_c(\boldsymbol{x}))_i) \right) - (\boldsymbol{f}_c(\boldsymbol{x}))_y$, where $\mathbf{1}(\cdot)$ is the indicator function. In practice, ERM is commonly used to compute an approximation of $R_c(\boldsymbol{f}_c)$ based on the client's local data $\mathcal{D}_c$ by $\widehat{R}_c(\boldsymbol{f}_c; \mathcal{D}_c) = \frac{1}{n_c} \sum_{i=1}^{n_c} \ell(\boldsymbol{f}_c(\boldsymbol{x}_i^c), y_i^c)$.

The goal of standard FL (McMahan et al., 2017) is that the $C$ clients collaboratively train a global classification model $\boldsymbol{f}$ that generalizes well with respect to $p(\boldsymbol{x}, y)$, without sharing their local data $\mathcal{D}_c$. The problem can be formalized as minimizing the aggregated risk:

$$R(\boldsymbol{f}) = \frac{1}{C} \sum_{c=1}^C R_c(\boldsymbol{f}). \tag{2}$$

Typically, we employ a server to coordinate the iterative distributed training as follows:

- in each global round of training, the server broadcasts its current model $\boldsymbol{f}$ to all the clients $\{\boldsymbol{f}_c\}_{c=1}^C$;
- each client $c$ copies the current server model $\boldsymbol{f}_c = \boldsymbol{f}$, performs $L$ local step updates

$$\boldsymbol{f}_c \leftarrow \boldsymbol{f}_c - \alpha_l \cdot \nabla \widehat{R}_c(\boldsymbol{f}_c; \mathcal{D}_c), \tag{3}$$

  where $\alpha_l$ is the local step-size, and sends $\boldsymbol{f}_c - \boldsymbol{f}$ back to the server;
- the server aggregates the updates $\{\boldsymbol{f}_c - \boldsymbol{f}\}_{c=1}^C$ to form a new server model using FedAvg:

$$\boldsymbol{f} \leftarrow \boldsymbol{f} - \alpha_g \cdot \sum_{c=1}^C (\boldsymbol{f}_c - \boldsymbol{f}), \tag{4}$$

  where $\alpha_g$ is the global step-size.

## 3  FEDERATED LEARNING: NO LABEL NO CRY

Next, we formulate the problem setting of FL with only U training sets, propose our method called *federation of unsupervised learning* (FedUL), and provide the theoretical analysis of FedUL. All proofs are given in Appendix C.

### 3.1  PROBLEM FORMULATION

In this paper, we consider a challenging setting: instead of fully labeled training data, each client $c \in [C]$ observes $M_c$ $(M_c \geq K, \forall c)$[1] sets of *unlabeled* samples, $\mathcal{U}_c = \{\mathcal{U}_c^m\}_{m=1}^{M_c}$, where $\mathcal{U}_c^m = $

---

[1]Note that we do not require each client has the same number of U sets. We only assume the condition $M_c \geq K, \forall c$ to ensure the full column rank matrix $\Pi_c$, which is essential for FL with U training sets.

$\{\boldsymbol{x}_i^{c,m}\}_{i=1}^{n_{c,m}}$ denotes the collection of the $m$-th U set for client $c$ with sample size $n_{c,m}$. Each U set can be seen as a set of data points drawn from a mixture of the original class-conditional densities:

$$\mathcal{U}_c^m \sim p_c^m(\boldsymbol{x}) = \sum_{k=1}^{K} \pi_c^{m,k} p(\boldsymbol{x} \mid y = k), \tag{5}$$

where $\pi_c^{m,k} = p_c^m(y = k)$ denotes the $k$-th class prior of the $m$-th U set at client $c$. These class priors form a full column rank matrix $\Pi_c \in \mathbb{R}^{M_c \times K}$ with the constraint that each row sums up to 1. Note that we do not require *any* labels to be known, only assume the class priors within each of the involved datasets are available, i.e., the training class priors $\pi_c^{m,k}$ and the test class prior $\pi^k = p(y = k)$.

Our goal is the same as standard FL: the $C$ clients are interested in collaboratively training a global classification model $\boldsymbol{f}$ that generalizes well with respect to $p(\boldsymbol{x}, y)$, but using only U training sets (5) on each local client.

## 3.2 PROPOSED METHOD

As discussed, FedAvg is a representative approach for aggregating clients in the FL framework. It also serves as the basis for many advanced federated aggregation techniques, e.g., Fedprox (Li et al., 2020), FedNova (Wang et al., 2020), and SCAFFOLD (Karimireddy et al., 2020). However, these methods cannot handle clients *without* labels. To tackle this problem, we consider the construction of *surrogate clients* by transforming the local U datasets into labeled datasets and modifying the local model such that it is compatible with the transformed data for training.

**Local Data Transformation**    First, we transform the data for the clients. For each client $c \in [C]$, let $\overline{y}$ be the index of given U sets, and $\overline{p}_c(\boldsymbol{x}, \overline{y})$ be an underlying joint distribution for the random variables $\boldsymbol{x} \in \mathcal{X}$ and $\overline{y} \in [M]$, where $M = \max_{c \in [C]} M_c$.[2] By treating $\overline{y}$ as a *surrogate label*, we may transform the original U training sets $\mathcal{U}_c = \{\mathcal{U}_c^m\}_{m=1}^{M_c}$ to a labeled training set for client $c$,

$$\overline{\mathcal{U}}_c = \{\boldsymbol{x}_i, \overline{y}_i\}_{i=1}^{n_c} \sim \overline{p}_c(\boldsymbol{x}, \overline{y}), \tag{6}$$

where $n_c = \sum_{m=1}^{M_c} n_{c,m}$. Obviously, the surrogate class-conditional density $\overline{p}_c(\boldsymbol{x} \mid \overline{y} = m)$ corresponds to $p_c^m(\boldsymbol{x})$ in (5), for $m \in [M_c]$. If $\exists M_c < m \le M$, we pad $\overline{p}_c(\boldsymbol{x} \mid \overline{y} = m) = 0$. The surrogate class prior $\overline{\pi}_c^m = \overline{p}_c(\overline{y} = m)$ can be estimated by $n_{c,m}/n_c$.

Based on the data transformation, we then formulate a *surrogate supervised FL task*: the $C$ clients aim to collaboratively train a global classification model $\boldsymbol{g} : \mathcal{X} \to \mathbb{R}^M$ that predicts the surrogate label $\overline{y}$ for the given input $\boldsymbol{x}$, without sharing their local data $\overline{\mathcal{U}}_c$. This task can be easily solved by standard supervised FL methods. More specifically, we minimize

$$J(\boldsymbol{g}) = \frac{1}{C} \sum_{c=1}^{C} J_c(\boldsymbol{g}) := \mathbb{E}_{(\boldsymbol{x}, \overline{y}) \sim \overline{p}_c(\boldsymbol{x}, \overline{y})}[\ell(\boldsymbol{g}(\boldsymbol{x}), \overline{y})] \tag{7}$$

by employing a server $\boldsymbol{g}$ that iteratively broadcasts its model to all local clients $\{\boldsymbol{g}_c\}_{c=1}^{C}$ and aggregates local updates from all clients using e.g., FedAvg. The local and global update procedures exactly follow (3) and (4) in standard FL.

Now a natural question arises: can we *infer* our desired model $\boldsymbol{f}$ from the surrogate model $\boldsymbol{g}$? To answer it, we study the relationship between the original and surrogate class-posterior probabilities.

**Theorem 1.** *For each client $c \in [C]$, let $\overline{\boldsymbol{\eta}}_c : \mathcal{X} \to \Delta_{M-1}$ and $\boldsymbol{\eta} : \mathcal{X} \to \Delta_{K-1}$ be the surrogate and original class-posterior probability functions, where $(\overline{\boldsymbol{\eta}}_c(\boldsymbol{x}))_m = \overline{p}_c(\overline{y} = m \mid \boldsymbol{x})$ for $m \in [M]$, $(\boldsymbol{\eta}(\boldsymbol{x}))_k = p(y = k \mid \boldsymbol{x})$ for $k \in [K]$, $\Delta_{M-1}$ and $\Delta_{K-1}$ are the $M$- and $K$-dimensional simplexes. Let $\overline{\boldsymbol{\pi}}_c = [\overline{\pi}_c^1, \cdots, \overline{\pi}_c^M]^\top$ and $\boldsymbol{\pi} = [\pi^1, \cdots, \pi^K]^\top$ be vector forms of the surrogate and original class priors, and $\Pi_c \in \mathbb{R}^{M \times K}$ be the matrix form of $\pi_c^{m,k}$ defined in (5). Then we have*

$$\overline{\boldsymbol{\eta}}_c(\boldsymbol{x}) = \boldsymbol{Q}_c(\boldsymbol{\eta}(\boldsymbol{x}); \boldsymbol{\pi}, \overline{\boldsymbol{\pi}}_c, \Pi_c), \tag{8}$$

*where the unnormalized version of vector-valued function $\boldsymbol{Q}_c$ is given by*

$$\widetilde{\boldsymbol{Q}}_c(\boldsymbol{\eta}(\boldsymbol{x}); \boldsymbol{\pi}, \overline{\boldsymbol{\pi}}_c, \Pi_c) = D_{\overline{\boldsymbol{\pi}}_c} \cdot \Pi_c \cdot D_{\boldsymbol{\pi}}^{-1} \cdot \boldsymbol{\eta}(\boldsymbol{x}).$$

*$D_{\boldsymbol{a}}$ denotes the diagonal matrix with diagonal terms being vector $\boldsymbol{a}$ and $\cdot$ denotes matrix multiplication. $\boldsymbol{Q}_c$ is normalized by the sum of all entries, i.e., $(\boldsymbol{Q}_c)_i = \frac{(\widetilde{\boldsymbol{Q}}_c)_i}{\sum_j (\widetilde{\boldsymbol{Q}}_c)_j}$.*

---

[2]To make the problem well-defined, for client $c$ with $M_c < M$, we set $\mathcal{U}_c^m = \emptyset, \forall M_c < m \le M$.

---

**Algorithm 1** Federation of unsupervised learning (FedUL)

---

**Server Input:** initial $\boldsymbol{f}$, global step-size $\alpha_g$, and global communication round $R$

**Client $c$'s Input:** local model $\boldsymbol{f}_c$, unlabeled training sets $\mathcal{U}_c = \{\mathcal{U}_c^m\}_{m=1}^{M_c}$, class priors $\Pi_c$ and $\boldsymbol{\pi}$, local step-size $\alpha_l$, and local updating iterations $L$

1: We start with initializing clients with **Procedure A**.
2: For $r = 1 \to R$ rounds, we run **Procedure B** and **Procedure C** iteratively .
3: **procedure A**. CLIENTINIT($c$)
4:     transform $\mathcal{U}_c$ to a surrogate labeled dataset $\overline{\mathcal{U}}_c$ according to (6)
5:     modify $\boldsymbol{f}$ to $\boldsymbol{g}_c = \boldsymbol{Q}_c(\boldsymbol{f})$, where $\boldsymbol{Q}_c$ is computed according to Theorem 1
6: **end procedure**
7: **procedure B**. CLIENTUPDATE($c$)
8:     $\boldsymbol{f}_c \leftarrow \boldsymbol{f}$                                      ▷ Receive updated model from SERVEREXECUTE
9:     $\boldsymbol{g}_c \leftarrow \boldsymbol{Q}_c(\boldsymbol{f}_c)$
10:    **for** $l = 1 \to L$ **do**
11:        $\boldsymbol{g}_c \leftarrow \boldsymbol{g}_c - \alpha_l \cdot \nabla \widehat{J}_c(\boldsymbol{g}_c; \overline{\mathcal{U}}_c)$              ▷ SGD update based on objective (7)
12:        $\boldsymbol{f}_c \leftarrow \boldsymbol{f}_c - \alpha_l \cdot \nabla \widehat{J}_c(\boldsymbol{Q}_c(\boldsymbol{f}_c); \overline{\mathcal{U}}_c)$        ▷ The update on $\boldsymbol{g}_c$ induces an update on $\boldsymbol{f}_c$
13:    **end for**
14:    send $\boldsymbol{f}_c - \boldsymbol{f}$ to SERVEREXECUTE
15: **end procedure**
16: **procedure C**. SERVEREXECUTE($r$)
17:    receive local models' updates $\{\boldsymbol{f}_c - \boldsymbol{f}\}_{c=1}^C$ from CLIENTUPDATE
18:    $\boldsymbol{f} \leftarrow \boldsymbol{f} - \alpha_g \cdot \sum_{c=1}^C (\boldsymbol{f}_c - \boldsymbol{f})$              ▷ FL aggregation
19:    broadcast $\boldsymbol{f}$ to CLIENTUPDATE
20: **end procedure**

---

Note that $\boldsymbol{\pi}$, $\overline{\boldsymbol{\pi}}_c$, and $\Pi_c$ are all fixed, $\boldsymbol{Q}_c$ is deterministic and can be identified with the knowledge of $\boldsymbol{\pi}$, $\Pi_c$, and an estimate of $\overline{\boldsymbol{\pi}}_c$. In addition, we note that if $M_c < M$, the last $M - M_c$ entries of $\boldsymbol{Q}_c$ are padded with zero by construction. We further study the property of $\boldsymbol{Q}_c$ in the following lemma.

**Lemma 2.** *For each client $c \in [C]$, the transition function $\boldsymbol{Q}_c : (\mathcal{X} \to \Delta_{K-1}) \to (\mathcal{X} \to \Delta_{M-1})$ defined in Theorem 1 is an injective function.*

In this sense, the transition function $\boldsymbol{Q}_c$ naturally bridges the class-posterior probability of our *desired* task $\boldsymbol{\eta}(\boldsymbol{x})$ and that of the *learnable* surrogate task $\overline{\boldsymbol{\eta}}_c(\boldsymbol{x})$ given only U training sets. This motivates us to embed the estimation of $\boldsymbol{\eta}(\boldsymbol{x})$ into the estimation of $\overline{\boldsymbol{\eta}}_c(\boldsymbol{x})$ at the client side.

**Local Model Modification**    Second, we modify the model for the clients. Let $\boldsymbol{f}(\boldsymbol{x})$ be the original global model output that estimates $\boldsymbol{\eta}(\boldsymbol{x})$. At each client $c \in [C]$, we implement $\boldsymbol{Q}_c$ by adding a transition layer following $\boldsymbol{f}_c$ and then we have $\boldsymbol{g}_c(\boldsymbol{x}) = \boldsymbol{Q}_c(\boldsymbol{f}_c(\boldsymbol{x}))$. Based on the local model modification, (7) can be reformulated as follows:

$$J(\boldsymbol{f}) = \frac{1}{C} \sum_{c=1}^C J_c(\boldsymbol{Q}_c(\boldsymbol{f})) := \mathbb{E}_{(\boldsymbol{x}, \overline{y}) \sim \overline{p}_c(\boldsymbol{x}, \overline{y})}[\ell(\boldsymbol{Q}_c(\boldsymbol{f}(\boldsymbol{x})), \overline{y})]. \tag{9}$$

Therefore, local updates on the surrogate model $\boldsymbol{g}_c$ naturally yield updates on the underlying model $\boldsymbol{f}$. Now we can aggregate the surrogate clients by a standard FL scheme. A detailed algorithm, using FedAvg as an example, is given in Algorithm 1.

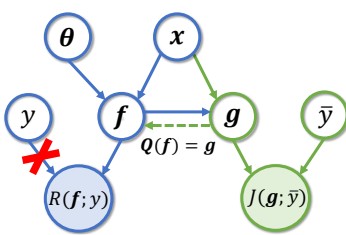

**Remarks**    The relationship elucidating the supervised and surrogate FL paradigms (for a single client) is visually shown in Figure 2. The blue part illustrates a supervised FL scheme: the global model is $\boldsymbol{f}(\boldsymbol{x})$ parameterized by $\boldsymbol{\theta}$; the loss is $R(\boldsymbol{f}; y)$ in (2) where labels $y$ are observed. However, we cannot observe $y$ and thus employ a surrogate FedUL scheme which is shown in green: the global model is $\boldsymbol{g}(\boldsymbol{x})$ directly modified from $\boldsymbol{f}$; the loss is $J(\boldsymbol{g}; \overline{y})$ in (7) where $\overline{y}$ are surrogate labels.

Figure 2: Graphical representation of supervised FL (in blue) and FedUL (in green).

### 3.3 THEORETICAL ANALYSIS

In what follows, we provide theoretical analysis on the convergence and optimal model recovery for the proposed method.

**Convergence of the Algorithm** In supervised FL, convergence analysis has been studied with regularity conditions (see Section 3.2.2 in Kairouz et al. (2019) for a comprehensive survey), e.g., the bounded gradient dissimilarity (Karimireddy et al., 2020) that associates the client model with the server model for non-IID data distributions.[3] In our analysis, we focus on the behaviour of our surrogate learning objective $J$ with the effect from transformation $\boldsymbol{Q}_c$.

**Proposition 3** (Theorem V in Karimireddy et al. (2020)). *Assume that $J(\boldsymbol{g})$ in (7) satisfies (A1)-(A3) in Appendix C.3. Denote $\boldsymbol{g}^* = \operatorname{argmin}_{\boldsymbol{g} \in \mathcal{G}} \widehat{J}(\boldsymbol{g})$, where $\widehat{J}(\boldsymbol{g}) = \frac{1}{C}\sum_{c=1}^{C} \widehat{J}_c(\boldsymbol{g}) := \frac{1}{n_c}\sum_{i=1}^{n_c} \ell(\boldsymbol{g}(\boldsymbol{x}_i), \overline{y}_i)$. Let the global step-size $\alpha_g \geq 1$ and local step-size $\alpha_l \leq \frac{1}{8(1+B^2)\beta L \alpha_g}$, FedUL given by Algorithm 1 is expected to have contracting gradient: with the initialized model $\boldsymbol{g}^0$, $F := J(\boldsymbol{g}^0) - J(\boldsymbol{g}^*)$ and constant $M = \sigma\sqrt{1 + \frac{C}{\alpha_g^2}}$, in $R$ rounds, the learned model $\boldsymbol{g}^R$ satisfies*

$$\mathbb{E}[\|\nabla J(\boldsymbol{g}^R)\|^2] \leq \mathcal{O}\left(\frac{\beta M \sqrt{F}}{\sqrt{RLC}} + \frac{\beta^{1/3}(FG)^{2/3}}{(R+1)^{2/3}} + \frac{\beta B^2 F}{R}\right).$$

Note that the proposition statement is w.r.t. the convergence of $\boldsymbol{g}$ learned from the surrogate task. By model construction, $\boldsymbol{g}$ is built from $\boldsymbol{Q}_c(\boldsymbol{f})$. By Lemma 2, the convergence of trained model $\boldsymbol{g}$ to the optimal model $\boldsymbol{g}^*$ in the function class $\mathcal{G}$ implies the convergence of the retrieved model $\boldsymbol{f}$ to the optimal model $\boldsymbol{f}^* = \operatorname{argmin}_{\boldsymbol{f} \in \mathcal{F}} \widehat{J}(\boldsymbol{f})$ in the corresponding function class $\mathcal{F}$ (induced by $\mathcal{G}$), where $\widehat{J}(\boldsymbol{f}) = \frac{1}{C}\sum_{c=1}^{C} \widehat{J}_c(\boldsymbol{Q}_c(\boldsymbol{f})) := \frac{1}{n_c}\sum_{i=1}^{n_c} \ell(\boldsymbol{Q}_c(\boldsymbol{f}(\boldsymbol{x}_i)), \overline{y}_i)$ is the empirical version of (9).

**Optimal Model Recovery** In our context, the model is optimal in the sense that the minimizer of the surrogate task $J(\boldsymbol{f}; \overline{y})$ coincide with the minimizer of the supervised task $R(\boldsymbol{f}; y)$, as if all the data have been labeled. Based on the convergence of surrogate training, we show that FedUL can recover this optimal model.

**Theorem 4.** *Assume that the cross-entropy loss is chosen for $\ell$, and the model used for $\boldsymbol{g}$ is very flexible, e.g., deep neural networks, so that $\boldsymbol{g}^{**} \in \mathcal{G}$ where $\boldsymbol{g}^{**} = \operatorname{argmin}_{\boldsymbol{g}} J(\boldsymbol{g}; \overline{y})$. Let $\boldsymbol{f}^* = \operatorname{argmin}_{\boldsymbol{f} \in \mathcal{F}} \widehat{J}(\boldsymbol{f})$ be the optimal classifier learned with FedUL by Algorithm 1, and $\boldsymbol{f}^\star = \operatorname{argmin}_{\boldsymbol{f}} R(\boldsymbol{f}; y)$ be the optimal classifier learned with supervised FL. By assumptions in Proposition 3, we have $\boldsymbol{f}^* = \boldsymbol{f}^\star$.*

## 4 EXPERIMENTS

In this section, we conduct experiments to validate the effectiveness of the proposed FedUL method under various testing scenarios. Compared with the baseline methods, FedUL consistently achieves superior results on both benchmark and real-world datasets. The detailed experimental settings and additional supportive results are presented in Appendix D & E.

### 4.1 BENCHMARK EXPERIMENTS

**Setup:** As proof-of-concepts, we first perform experiments on widely adopted benchmarks MNIST (LeCun et al., 1998) and CIFAR10 (Krizhevsky et al., 2009). We generate $M$ sets of U training data according to (5) for each client $c \in [C]$. The total number of training samples $\overline{n}_c$ on each client is fixed, and the number of instances contained in each set indexed by $m$ is also fixed as $\overline{n}_c/M$. In order to simulate the real world cases, we uniformly select class priors $\pi_c^{m,k}$ from range $[0.1, 0.9]$ and then regularize them to formulate a valid $\Pi_c \in \mathbb{R}^{M \times K}$ as discussed in Section 3.1.

For model training, we use the cross-entropy loss and Adam (Kingma & Ba, 2015) optimizer with a learning rate of $1e-4$ and train 100 rounds. If not specified, our default setting for local update

---

[3]The IID case is equivalent to batch sampling from the same distribution, where the analysis is less interesting associated with our FL setting and we omit details here.

Table 1: Quantitative comparison of our method with the baseline methods on benchmark datasets under IID and Non-IID setting (mean error (std)). The best method (paired t-test at significance level 5%) is highlighted in boldface.

**IID Task with 5 Clients**

| Dataset | Sets | FedPL | FedLLP | FedLLP-VAT | FedUL | FedAvg 10% |
|---------|------|-------|--------|-----------|-------|-----------|
| MNIST   | 10   | 2.12 (0.10)  | 19.00 (15.90) | 20.53 (5.39)  | **0.78 (0.06)**  |             |
|         | 20   | 2.98 (0.15)  | 4.38 (3.95)   | 4.40 (5.11)   | **1.12 (0.09)**  | 1.79 (0.09) |
|         | 40   | 3.90 (0.22)  | 10.11 (15.74) | 11.96 (9.56)  | **1.00 (0.29)**  |             |
| CIFAR10 | 10   | 34.60 (0.30) | 77.01 (5.02)  | 77.56 (8.71)  | **18.56 (0.04)** |             |
|         | 20   | 42.06 (1.63) | 78.74 (6.96)  | 84.18 (5.95)  | **19.66 (0.30)** | 32.85 (1.03)|
|         | 40   | 46.20 (0.67) | 73.75 (4.78)  | 78.65 (6.91)  | **20.31 (0.69)** |             |

**Non-IID Task with 5 Clients**

| Dataset | Sets | FedPL | FedLLP | FedLLP-VAT | FedUL | FedAvg 10% |
|---------|------|-------|--------|-----------|-------|-----------|
| MNIST   | 10   | 15.54 (3.48) | 20.52 (15.00) | 23.39 (15.34) | **2.98 (1.72)**  |             |
|         | 20   | 7.54 (2.77)  | 6.63 (7.94)   | 11.76 (6.60)  | **1.77 (0.49)**  | 3.82 (1.56) |
|         | 40   | 5.76 (0.90)  | 5.53 (2.51)   | 7.82 (10.73)  | **1.64 (0.18)**  |             |
| CIFAR10 | 10   | 70.65 (2.38) | 81.12 (5.41)  | 81.55 (7.09)  | **42.92 (3.02)** |             |
|         | 20   | 63.53 (1.24) | 65.51 (13.16) | 70.51 (7.66)  | **35.24 (0.98)** | 58.62 (8.63)|
|         | 40   | 53.78 (0.13) | 65.72 (3.93)  | 60.23 (5.71)  | **31.43 (1.72)** |             |

epochs (E) is 1 and batch size (B) is 128 with 5 clients (C) in our FL system. We use LeNet (LeCun et al., 1998) and ResNet18 (He et al., 2016) as backbone for MNIST and CIFAR10, respectively. Here, we investigate two different FL settings: **IID** and **Non-IID**, where the overall label distribution across clients is the same in the IID setting, whereas clients have different label distributions in the non-IID setting. Specially, for Non-IID setting, we impose class-prior shift as follows:

- the classes are divided into the majority and minority classes, where the fraction of each minority class is less than $0.08$ while the fraction of each majority class is in the range $[0.15, 0.25]$;
- the training data for every client are sampled from 2 majority classes and 8 minority classes;
- the test data are uniformly sampled from all classes.

**Baselines:** (1) Federated Pseudo Labeling (FedPL) (Diao et al., 2021): at each client of a FedAvg system, we choose the label with the largest class prior in a U set as the pseudo label for all the data in this set. During local training, these pseudo labels are boosted by *Mixup* (Zhang et al., 2017) high-confidence data and low-confidence data, where the confidence is given by model predictions. (2) Federated LLP (FedLLP) (Dulac-Arnold et al., 2019): each client model is updated by minimizing the distance between the ground truth label proportion and the predicted label proportion, and we aggregate them using FedAvg. (3) FedLLP with Virtual Adversarial Training (FedLLP-VAT) (Tsai & Lin, 2020): a consistency term is added to FedLLP as regularization for optimizing client model in FedAvg. (4) FedAvg with 10% labeled data (FedAvg 10%): following Chen et al. (2020), we also compare with supervised learning based on FedAvg using 10% fully labeled data. The detailed baseline settings are depicted in Appendix D.

**Results:** The experimental results of our method and baselines on two benchmark datasets under IID and Non-IID settings are reported in Table 1, where the mean error (std) over clients based on 3 independent runs are shown. Given the limited number of total data available in the benchmark datasets, we compare our method with baselines in an FL system with five clients and vary the number of U sets $M \in \{10, 20, 40\}$ at each client. In this regard, each set contains a sufficient number of data samples to train the model.

Our observations are as follows. First, the proposed FedUL method outperforms other baseline methods by a large margin under both IID and Non-IID settings. Second, when varying the number of sets $M \in \{10, 20, 40\}$, FedUL not only achieves lower classification error but also shows more stable performance compared to baselines. Third, as we use FedAvg as the default aggregation strategy for all the methods, the performances of FedUL, FedPL, and FedAvg all degrade under the inter-client Non-IID setting, as expected. The error rates of FedLLP and FedLLP-VAT are high and unstable, which demonstrates that the EPRM based methods are inferior as discussed in Section 1.

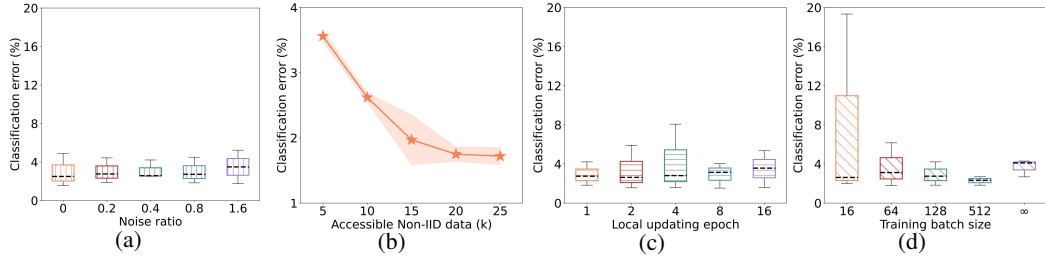

Figure 3: Ablation studies on MNIST under a Non-IID FL system. (a) Study on the robustness of our method with noisy class priors. (b) Effects of the number of accessible Non-IID U data. (c) Analysis of local updating epochs. (d) Effects of training batch size.

## 4.2 ABLATION STUDIES

In the following, we present a comprehensive investigation on the ablation studies of the proposed FedUL method on MNIST benchmark.

**Robustness with Noisy Class Priors:** To investigate the robustness of our method, we further apply FedUL in a noisy environment. Specifically, we perturb the class priors $\pi_c^{m,k}$ by a random noise $\gamma$, and let the method treat the noisy class prior $\widetilde{\pi}_c^{m,k} = \pi_c^{m,k} \cdot ((2\epsilon - 1)\gamma + 1)$, where $\gamma$ uniform randomly take values in $[0, 1]$ and $\epsilon \in \{0, 0.2, 0.4, 0.8, 1.6\}$, as the true $\pi_c^{m,k}$ during the whole learning process. Note that we tailor the noisy $\widetilde{\pi}_c^{m,k}$ if it surpasses the range. We conduct experiments with five clients and ten U sets at each client. The experimental setup is the same as before except for the replacement of $\pi_c^{m,k}$. From Figure 3(a), we can see that FedUL performs stably even with some challenging noise perturbations (see when noise ratio $\epsilon > 0$).

**Effects of the Number of Accessible Non-IID U Data:** The purpose of FL is utilizing more data to empower model performance. While how unlabeled data, especially unlabeled Non-IID data, can contribute to FL model performance needs investigation. To this end, we analyze the effects of the number of accessible Non-IID U data. We set each client with $n_c = 500$ samples $\forall c \in [C]$ and conduct experiments by changing the amount of Non-IID U data by increasing the number of clients $C \in \{10, 20, 30, 40, 50\}$. As shown in Figure 3(b), the model performance improves by a large margin as the number of accessible Non-IID U data increases by using our proposed FedUL method.

**Analysis of Local Updating Epochs:** The frequency of aggregation may sometimes influence the model performance in an FL system. We would like to explore if FedUL is stable when the local clients are updated with different local epochs $E$. In Figure 3(c), we explore $E \in \{1, 2, 4, 8, 16\}$ with fixed $B = 128$ and $C = 5$. The results demonstrate that with different $E$, the performances of FedUL are reasonably stable.

**Effects of Batch Size:** The training batch size is always an important factor in machine learning tasks, especially in an FL system. Hence here, we study how batch size affects our model performance. Specifically, we train the model using different batch sizes $B \in \{16, 64, 128, 512, \infty\}$ with fixed $E = 1$ and $C = 5$. Here $B = \infty$ stands for loading all of the training samples into a single batch. The results in Figure 3(d) indicate that FedUL can effectively work under a wide range of batch sizes.

## 4.3 CASE STUDY ON REAL-WORLD PROBLEM

To demonstrate the feasibility of our proposed FedUL scheme to the real problem, we test FedUL on a medical application for the diagnosis of brain disease. In the rest of this section, we introduce the problem formulation, data and setting, and then discuss the results.

**Real Problem Formulation:** For the real-world scenario, we consider a clinical setting for disease classification in $C$ different healthcare systems. Suppose each healthcare system has $M$ hospitals located in different regions, each associated with unique class priors due to population diversities. Medical data are not shareable between the healthcare systems for privacy reasons. We assume patient-wise diagnosis information is unavailable, but each hospital provides the prevalence of these patients (i.e., class priors of the disease classes).

Table 2: Quantitative comparison of our method with the baseline methods for the five healthcare systems (HS) using the real-world intracranial hemorrhage detection dataset. Results of three runs are reported in mean error (std) format. The best methods are highlighted in boldface.

| Hospitals | Method | HS 1 | HS 2 | HS 3 | HS 4 | HS 5 |
|---|---|---|---|---|---|---|
| 12 | FedPL | 44.69 (4.28) | 46.42 (4.35) | 46.29 (1.83) | 47.55 (2.84) | 46.53 (2.40) |
| | **FedUL (ours)** | **40.26 (1.39)** | **40.35 (1.10)** | **39.31 (1.10)** | **39.72 (0.67)** | **39.43 (1.42)** |
| 24 | FedPL | 42.71 (2.97) | 42.18 (3.00) | 42.30 (1.28) | 41.79 (1.98) | 41.46 (0.53) |
| | **FedUL (ours)** | **32.27 (2.71)** | **30.99 (2.03)** | **31.29 (1.05)** | **31.72 (1.23)** | **31.37 (1.58)** |
| 48 | FedPL | 41.44 (3.04) | 40.38 (1.38) | 41.06 (2.88) | 41.44 (1.78) | 40.26 (2.35) |
| | **FedUL (ours)** | **31.48 (2.25)** | **30.92 (0.88)** | **31.16 (0.84)** | **31.55 (0.58)** | **30.57 (1.37)** |
| - | FedAvg 10% | 46.79 (1.91) | 46.44 (0.78) | 47.19 (1.05) | 47.29 (1.93) | 46.20 (1.61) |

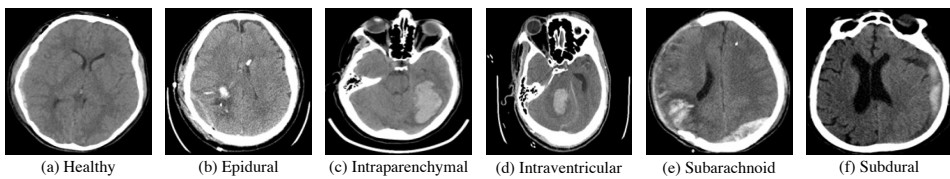

(a) Healthy     (b) Epidural     (c) Intraparenchymal     (d) Intraventricular     (e) Subarachnoid     (f) Subdural

Figure 4: Example samples of each class in the distributed medical dataset (Flanders et al., 2020).

**Dataset and Settings:** We use the RSNA Intracranial Hemorrhage Detection dataset (Flanders et al., 2020), which contains large-scale Magnetic Response Image (MRI) data from several hospitals (see Figure 4), to simulate this real-world scenario. There are $106k$ samples in the dataset belonging to six classes, including five types of intracranial hemorrhage (Epidural, Intraparenchymal, Intraventricular, Subarachnoid, and Subdural) and healthy control type. We model an FL system with five healthcare systems (i.e., clients). Each healthcare system contains $21.2k$ data samples. To show the effect on the number of samples in each U set, we vary the number of hospitals (i.e., sets) $M = 12, 24$, and $48$ in each system. Here we generate each $\pi_c^{m,k}$ for the $k$-th class in $m$-th U set at $c$-th client from range $[0.1, 0.9]$ and check that the generated class priors are not all identical. For model training, we use the cross-entropy loss and Adam (Kingma & Ba, 2015) optimizer with a learning rate of $1e - 4$ and train 100 rounds. We use ResNet18 (He et al., 2016) as backbone for this experiment.

**Results and Analysis:** As indicated in our proof-of-concept (Table 1), FedPL consistently outperforms the other two LLP baselines, i.e., FedLLP and FedLLP-VAT. Here we focus on comparing FedUL with FedPL. The supervised baseline FedAvg with 10% fully labeled data is also included for reference. We report the results of each hospital in mean error (std) format over three runs in Table 2. We can see that our proposed FedUL method consistently outperforms FedPL and FedAvg 10% for all the hospitals and under different set number settings. The superiority of FedUL in this challenging real-world FL setting further indicates the effectiveness and robustness of our algorithm. These results are inspiring and bring the hope of deploying FedUL to the healthcare field, where data are often very costly to be labeled, heterogeneous, and distributed.

## 5 CONCLUSION

In this paper, we propose FedUL, a novel FL algorithm using only U data. FedUL works as a wrapper that transforms the unlabeled client data and modifies the client model to form a surrogate supervised FL task, which can be easily solved using existing FL paradigms. Theoretically, we prove that the recovery of the optimal model by using the FedUL method can be guaranteed under mild conditions. Through extensive experiments on benchmark and real-world datasets, we show FedUL can significantly outperform the baseline methods under different settings. As a wrapper, we expect FedUL can be further improved by incorporating advanced FL aggregation or optimization schemes for the non-IID setting (Wang et al., 2020; Li et al., 2020; 2021; Karimireddy et al., 2020).

## ETHICS STATEMENT

Collecting sample-wise labels for training can be costly and may violate data privacy. Considering data in the real world are often distributed at a large scale, we provide a promising solution to perform multi-class classification on distributed U data in the FL system. Our method FedUL can be widely applied to data-sensitive fields, such as healthcare, politics, and finance, where the sample-wise label is not available, but populational prevalence can be accessed. With FedUL, we hope the future FL can enjoy significant labeling cost reduction and improve its privacy. The real datasets used in this work are publicly available with appropriate prepossessing to de-identify patients' identities. We provide proper references to the datasets, libraries, and tools used in our study. This study does not have any legal compliance, conflict of interest, or research integrity issues.

## REPRODUCIBILITY STATEMENT

We provide the codes to reproduce the main experimental results at https://github.com/lunanbit/FedUL. We further provide experimental details and additional information of the datasets in the supplementary materials.

### ACKNOWLEDGMENTS

NL was supported by JST AIP Challenge Program and the Institute for AI and Beyond, UTokyo. ZW and QD were supported by project of Shenzhen-HK Collaborative Development Zone. XL was thankful for the support by NVIDIA. GN was supported by JST AIP Acceleration Research Grant Number JPMJCR20U. MS was supported by JST AIP Acceleration Research Grant Number JPMJCR20U3 and the Institute for AI and Beyond, UTokyo.

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

# Supplementary Material

**Roadmap**  The appendix is organized as follows. We summarize notations used in our paper in Section A and give a full discussion of related work in Section B. Our proofs are presented in Section C. We provide detailed experimental settings in Section D and supplementary experimental results on the benchmark datasets in Section E.

## A  NOTATION TABLE

Table A.1: Notations used in the paper.

| Notations | Description |
|---|---|
| $d$ | input dimension |
| $K$ | number of classes for the original classification task |
| $k$ | index of the class $k \in [K]$ |
| $\mathcal{X}$ | input space $\mathcal{X} \subset \mathbb{R}^d$ |
| $\mathcal{Y}$ | label space $\mathcal{Y} = [K]$ where $[K] := \{1, \dots, K\}$ |
| $\boldsymbol{x}$ | input $\boldsymbol{x} \in \mathcal{X}$ |
| $y$ | true label $y \in \mathcal{Y}$ |
| $p$ | test data distribution |
| $\pi^k$ | $k$-th class prior $\pi^k := p(y = k)$ of the test set |
| $\boldsymbol{\pi}$ | vector form of $\pi^k$ where $\boldsymbol{\pi} = [\pi^1, \cdots, \pi^K]^\top$ |
| $\ell$ | loss function for a classification task |
| $R$ | risk for the original FL task |
| $\boldsymbol{f}$ | global model predicting label $y$ for input $\boldsymbol{x}$ |
| $C$ | number of clients in the FL setup |
| $c$ | index of the client $c \in [C]$ |
| $\boldsymbol{f}_c$ | $c$-th client model predicting label $y$ for input $\boldsymbol{x}$ |
| $p_c$ | local data distribution at the $c$-th client |
| $n_c$ | number of training data at the $c$-th client |
| $\mathcal{D}_c$ | labeled training set $\mathcal{D}_c = \{(\boldsymbol{x}_i^c, y_i^c)\}_{i=1}^{n_c}$ at the $c$-th client |
| $\alpha_l$ | local step size |
| $\alpha_g$ | global step size |
| $R$ | global communication round |
| $L$ | local updating iterations |
| $M_c$ | number of available U sets at the $c$-th client |
| $m$ | index of the U set $m \in [M_c]$ |
| $n_{c,m}$ | number of training data in the $m$-th U set at the $c$-th client |
| $p_c^m$ | $m$-th U data distribution at the $c$-th client |
| $\mathcal{U}_c^m$ | $m$-th U set at the $c$-th client $\mathcal{U}_c^m = \{\boldsymbol{x}_i^{c,m}\}_{i=1}^{n_{c,m}}$ |
| $\mathcal{U}_c$ | all available U sets at the $c$-th client $\mathcal{U}_c = \{\mathcal{U}_c^m\}_{m=1}^{M_c}$ |
| $M$ | the maximum number of available U sets among all clients $M = \max_{c \in [C]} M_c$ |
| $\pi_c^{m,k}$ | $k$-th class prior $\pi_c^{m,k} = p_c^m(y = k)$ of the $m$-th U set at the $c$-th client |
| $\Pi_c$ | matrix form of $\pi_c^{m,k}$ where $\Pi_c \in \mathbb{R}^{M_c \times K}$ |
| $\overline{y}$ | surrogate label $\overline{y} \in [M]$ |
| $\overline{p}_c$ | surrogate local data distribution at the $c$-th client |
| $\overline{\mathcal{U}}_c$ | surrogate labeled training set $\overline{\mathcal{U}}_c = \{\boldsymbol{x}_i, \overline{y}_i\}_{i=1}^{n_c}$ at the $c$-th client |
| $\overline{\pi}_c^m$ | $m$-th surrogate class prior $\overline{\pi}_c^m = \overline{p}_c(\overline{y} = m)$ at the $c$-th client |
| $\overline{\boldsymbol{\pi}}_c$ | vector form of $\overline{\pi}_c^m$ where $\overline{\boldsymbol{\pi}}_c = [\overline{\pi}_c^1, \cdots, \overline{\pi}_c^M]^\top$ |
| $\boldsymbol{g}$ | surrogate global model predicting surrogate label $\overline{y}$ for input $\boldsymbol{x}$ |
| $\overline{\boldsymbol{\eta}}_c$ | surrogate class-posterior probability at the $c$-th client where $(\overline{\boldsymbol{\eta}}_c(\boldsymbol{x}))_m = \overline{p}_c(\overline{y} = m \mid \boldsymbol{x})$ |
| $\boldsymbol{\eta}$ | true class-posterior probability at the $c$-th client where $(\boldsymbol{\eta}(\boldsymbol{x}))_k = p(y = k \mid \boldsymbol{x})$ |
| $\boldsymbol{Q}_c$ | vector-valued transition function $\boldsymbol{Q}_c : (\mathcal{X} \to \Delta_K) \to (\mathcal{X} \to \Delta_M)$ |
| $J$ | surrogate risk for the surrogate FL task |

# B  RELATED WORK

## B.1  PERSONALIZED FEDERATED LEARNING

Considering the heterogeneity of data, clients, and systems, personalization is a natural and trending solution to improve FL performance. Many efforts have been made in personalized FL. FedProx(Li et al., 2020), FedCurv (Shoham et al., 2019), and FedCL (Yao & Sun, 2020) propose regularization-based methods that include special regularization terms to improve FL model generalizing to various clients. Meta-learning methods, such as MAML (Finn et al., 2017), are also used to learn personalized models for each client (Jiang et al., 2019; Fallah et al., 2020; Li et al., 2019). Unlike the methods mentioned above that yield a single generalizable model, multiple FL models can be trained using clustering strategies (Sattler et al., 2020; Ghosh et al., 2020). Different local FL models can also be achieved by changing the last personalized layers (Li et al., 2021; Arivazhagan et al., 2019).

At first glance, FedUL looks similar to personalized FL since each client learns a "personalized" model $g_c$ which contains a client-specific transition layer $Q_c$. However, our learning setting and method are very different from existing personalized FL approaches, as explained below. First, our framework focuses on FL with only unlabeled data. The personalized model $g_c$ only serves as a proxy, in order to be compatible with the transformed unlabeled data, and our ultimate goal is to learn $f$ that is not exactly personalized. Specifically, 1) if the inter-clients are IID, FedUL yields the same proxy model $g_c$ and federated model $f$ for all the clients; 2) if the inter-clients are Non-IID, although $g_c$ is different (as $Q_c$ is different), the underlying model $f$ (what we want) is still the same. Second, our method differs from the previous personalized FL methods by its surrogate training nature: it works as a wrapper that transforms unlabeled clients to (surrogate) labeled ones and is compatible with many supervised FL methods.

## B.2  SEMI-SUPERVISED FEDERATED LEARNING

Another line of research working on utilizing unlabeled data in FL follows the semi-supervised learning framework (Zhu, 2005; Chapelle et al., 2006), which assumes a small amount of labeled data available at the client side (labels-at-client) or at the server side (labels-at-server). For example, FedMatch (Jeong et al., 2021) introduces a new inter-client consistency loss that maximizes the agreement between multiple clients and decomposes the model parameters into one for labeled data and the other for unlabeled data for disjoint learning. FedSem (Albaseer et al., 2020) generates pseudo labels for unlabeled data based on the trained FedAvg model with labeled data, and the generated (pseudo) labeled data are further used to retrain FedAvg to obtain the global model. Moreover, distillation-based semi-supervised FL (Itahara et al., 2020) considers using shared unlabeled data for distillation-based message exchanging.

Although the motivation of FedUL can be related to semi-supervised FL since both work on utilizing unlabeled data in FL, our learning setting is different since we only have unlabeled data at both the client and the server sides, and our method is different since FedUL follows the empirical risk minimization framework.

## C  PROOFS

In this appendix, we prove all theorems.

### C.1  PROOF OF THEOREM 1

By the Bayes' rule, $\forall m \in [M]$ we have:

$$
\begin{aligned}
(\overline{\boldsymbol{\eta}}_c(\boldsymbol{x}))_m &= \frac{\overline{p}_c(\boldsymbol{x}, \overline{y} = m)}{\overline{p}_c(\boldsymbol{x})} \\
&= \frac{\overline{p}_c(\boldsymbol{x}, \overline{y} = m)}{\sum_{m=1}^{M_c} \overline{p}_c(\boldsymbol{x}, \overline{y} = m)} \\
&= \frac{\overline{p}_c(\boldsymbol{x} \mid \overline{y} = m)\overline{p}_c(\overline{y} = m)}{\sum_{m=1}^{M_c} \overline{p}_c(\boldsymbol{x} \mid \overline{y} = m)\overline{p}_c(\overline{y} = m)} \\
&= \frac{p_c^m(\boldsymbol{x})\overline{\pi}_c^m}{\sum_{m=1}^{M_c} p_c^m(\boldsymbol{x})\overline{\pi}_c^m} \\
&= \frac{\overline{\pi}_c^m \sum_{k=1}^K \pi_c^{m,k} p(\boldsymbol{x} \mid y = k)}{\sum_{m=1}^{M_c} \overline{\pi}_c^m \sum_{k=1}^K \pi_c^{m,k} p(\boldsymbol{x} \mid y = k)} \\
&= \frac{\overline{\pi}_c^m \sum_{k=1}^K \pi_c^{m,k} \frac{(\boldsymbol{\eta}(\boldsymbol{x}))_k}{\pi^k}}{\sum_{m=1}^{M_c} \overline{\pi}_c^m \sum_{k=1}^K \pi_c^{m,k} \frac{(\boldsymbol{\eta}(\boldsymbol{x}))_k}{\pi^k}}.
\end{aligned}
\tag{10}
$$

By transforming (10) to the matrix form, we complete the proof. $\qquad\square$

### C.2  PROOF OF LEMMA 2

Given our data generation process in Section 3.1, we have that $M_c \geq k$ and $\Pi_c \in \mathbb{R}^{M_c \times K}$ is a full column rank matrix. With $D_{\overline{\boldsymbol{\pi}}_c}$ and $D_{\boldsymbol{\pi}}^{-1}$ being non-degenerate, we obtain that $\boldsymbol{Q}_c(\boldsymbol{\eta}(\boldsymbol{x}))$ presents an injective mapping from $\boldsymbol{\eta}(\boldsymbol{x})$ to $\overline{\boldsymbol{\eta}}_c(\boldsymbol{x})$. $\qquad\square$

### C.3  PROOF OF PROPOSITION 3

The assumptions **(A1)**-**(A3)** are listed as follows.

**(A1)** Bounded gradient dissimilarity: $\exists$ constants $G \geq 0$ and $B \geq 1$ s.t.
$$
\frac{1}{C} \sum_{c=1}^C \|\nabla J_c(\boldsymbol{g})\|^2 \leq G^2 + B^2 \|\nabla J(\boldsymbol{g})\|^2, \forall \boldsymbol{g}.
$$

**(A2)** $\beta$-smooth function $J_c$ satisfies
$$
\|\nabla J_c(\boldsymbol{\theta}) - \nabla J_c(\boldsymbol{\theta}')\| \leq \beta \|\boldsymbol{\theta} - \boldsymbol{\theta}'\|, \forall c, \boldsymbol{\theta}, \boldsymbol{\theta}'.
$$

**(A3)** Unbiased stochastic gradient with bounded variance:
$$
\mathbb{E}[\|\nabla \widehat{J}_c(\boldsymbol{g}; \overline{\mathcal{U}}_c) - \nabla J_c(\boldsymbol{g})\|^2] \leq \sigma^2, \forall c, \boldsymbol{g},
$$
where $\widehat{J}_c(\boldsymbol{g}; \overline{\mathcal{U}}_c) = \frac{1}{n_c} \sum_{i=1}^{n_c} \ell(\boldsymbol{g}(\boldsymbol{x}_i), \overline{y}_i)$.

Note that this proposition statement is w.r.t. the convergence of model $\boldsymbol{g}$ learned from the surrogate supervised learning task. The proof follows directly from Theorem V in Karimireddy et al. (2020). $\qquad\square$

### C.4  PROOF OF THEOREM 4

When a proper loss (Reid & Williamson, 2010) is used for $\ell$, the mapping of $\boldsymbol{g}^{**}$ is the unique minimizer of $J(\boldsymbol{g})$. We show an example of the a widely used proper loss function: the cross-entropy loss (De Boer et al., 2005) defined as

$$
\ell_{\mathrm{ce}}(\boldsymbol{g}(\boldsymbol{x}), \overline{y}) = -\sum_{m=1}^M \mathbf{1}(\overline{y} = m) \log(\boldsymbol{g}(\boldsymbol{x}))_m = -\log(\boldsymbol{g}(\boldsymbol{x}))_{\overline{y}}.
$$

We can see that the cross-entropy loss is non-negative by its definition. Therefore, minimizing $J(\boldsymbol{g})$ can be obtained by minimizing the conditional risk $\mathbb{E}_{p(\overline{y}|\boldsymbol{x})}[\ell(\boldsymbol{g}(\boldsymbol{x}), \overline{y}) \mid \boldsymbol{x}]$ for every $\boldsymbol{x} \in \mathcal{X}$. So we are now optimizing

$$\phi(\boldsymbol{g}) = -\sum_{m=1}^{M} p(\overline{y} = m \mid \boldsymbol{x}) \cdot \log(g(\boldsymbol{x}))_m, \quad \text{s.t.} \sum_{m=1}^{M} (g(\boldsymbol{x}))_m = 1.$$

By using the Lagrange multiplier method (Bertsekas, 1997), we have

$$\mathcal{L} = -\sum_{m=1}^{M} p(\overline{y} = m \mid \boldsymbol{x}) \cdot \log(g(\boldsymbol{x}))_m - \lambda \cdot \left(\sum_{m=1}^{M} (g(\boldsymbol{x}))_m - 1\right).$$

The derivative of $\mathcal{L}$ with respect to $\boldsymbol{g}$ is

$$\frac{\partial \mathcal{L}}{\partial \boldsymbol{g}} = \left[-\frac{p(\overline{y} = 1 \mid \boldsymbol{x})}{(\boldsymbol{g}(\boldsymbol{x}))_1} - \lambda, \cdots, -\frac{p(\overline{y} = M \mid \boldsymbol{x})}{(\boldsymbol{g}(\boldsymbol{x}))_M} - \lambda\right]^{\top}.$$

By setting this derivative to 0 we obtain

$$(\boldsymbol{g}(\boldsymbol{x}))_m = -\frac{1}{\lambda} \cdot p(\overline{y} = m \mid \boldsymbol{x}), \quad \forall m = 1, \ldots, M \text{ and } \forall \boldsymbol{x} \in \mathcal{X}.$$

Since $\boldsymbol{g} \in \Delta_{M-1}$ is the $M$-dimensional simplex, we have $\sum_{m=1}^{M} (\boldsymbol{g}^{**}(\boldsymbol{x}))_m = 1$ and $\sum_{m=1}^{M} p(\overline{y} = j \mid \boldsymbol{x}) = 1$. Then

$$\sum_{m=1}^{M} (\boldsymbol{g}^{**}(\boldsymbol{x}))_m = -\frac{1}{\lambda} \cdot \sum_{m=1}^{M} p(\overline{y} = m \mid \boldsymbol{x}) = 1.$$

Therefore we obtain $\lambda = -1$ and $(\boldsymbol{g}^{**}(\boldsymbol{x}))_m = p(\overline{y} = m \mid \boldsymbol{x}) = (\overline{\boldsymbol{\eta}}(\boldsymbol{x}))_m, \forall m = 1, \ldots, M$ and $\forall \boldsymbol{x} \in \mathcal{X}$, which is equivalent to $\boldsymbol{g}^{**}(\boldsymbol{x}) = \overline{\boldsymbol{\eta}}(\boldsymbol{x})$. Similarly, we obtain

$$\boldsymbol{f}^{\star}(\boldsymbol{x}) = \boldsymbol{\eta}(\boldsymbol{x}). \tag{11}$$

Given Proposition 3 and the assumption that the model used for $\boldsymbol{g}$ is sufficiently flexible, we obtain that the classification model learned by FedUL (see Algorithm 1) satisfies $\boldsymbol{g}^* = \boldsymbol{g}^{**}$. By our model construction, $\boldsymbol{g}$ is built from $\boldsymbol{Q}_c(\boldsymbol{f})$. Then we have

$$\boldsymbol{f}^* = \boldsymbol{f}^{**} \tag{12}$$

where $\boldsymbol{f}^{**} = \operatorname{argmin}_{\boldsymbol{f}} J(\boldsymbol{f})$ is the model induced by $\boldsymbol{g}^{**}$. Given Theorem 1 and Lemma 2, we obtain that $\boldsymbol{g}^{**}(\boldsymbol{x}) = \overline{\boldsymbol{\eta}}(\boldsymbol{x})$ if and only if

$$\boldsymbol{f}^{**}(\boldsymbol{x}) = \boldsymbol{\eta}(\boldsymbol{x}). \tag{13}$$

By summarizing (11), (12), and (13) we obtain $\boldsymbol{f}^* = \boldsymbol{f}^{\star}$. $\qquad\square$

## D    EXPERIMENTAL DETAILS

We illustrate our model architectures and training details of the benchmark and real-world experiments in this section.

### D.1    MODEL ARCHITECTURE

For our benchmark experiment on MNIST, we use a six-layer convolutional neural network. The network details are listed in Table D.2. For our benchmark experiment on CIFAR10 and real-world experiment, we use ResNet18 as our backbone. The network details are listed in Table D.3.

| Layer | Details |
|---|---|
| 1 | Conv2D(3, 64, 5, 1, 2), BN(64), ReLU, MaxPool2D(2, 2) |
| 2 | Conv2D(64, 64, 5, 1, 2), BN(64), ReLU, MaxPool2D(2, 2) |
| 3 | Conv2D(64, 128, 5, 1, 2), BN(128), ReLU |
| 4 | FC(6272, 2048), BN(2048), ReLU |
| 5 | FC(2048, 512), BN(512), ReLU |
| 6 | FC(512, 10) |

Table D.2: Model architecture of the benchmark experiment on MNIST. For convolutional layer (Conv2D), we list parameters with sequence of input and output dimension, kernel size, stride and padding. For max pooling layer (MaxPool2D), we list kernel and stride. For fully connected layer (FC), we list input and output dimension. For BatchNormalization layer (BN), we list the channel dimension.

### D.2    TRAINING DETAILS

**Benchmark Experiments:**    We implement all the methods by PyTorch and conduct all the experiments on an NVIDIA TITAN X GPU. Please note for all experiments, we split 20% original data for validation and model selection. During training process, we use Adam optimizer with learning rate $1e-4$ and the cross-entropy loss. We set batch size to 128 and training rounds to 100. To avoid overfitting, we use L1 regularization. The detailed weight of L1 regularization used in benchmark experiments is shown in Table D.4. The detailed number of samples at each client for MNIST and CIFAR10 under IID and Non-IID setting is given in Table D.5 and Table D.6, respectively.

We describe details of the baseline methods and corresponding hyper-parameters as follows.

- FedPL (Diao et al., 2021): this method is based on empirical classification risk minimization. The learning objective is

$$\widehat{R}(\boldsymbol{f}) = \frac{1}{C} \sum_{c=1}^{C} \sum_{m=1}^{M_c} \mathcal{L}_{fix} + \lambda \mathcal{L}_{mix}, \tag{14}$$

where

$$\mathcal{L}_{\text{fix}} = \ell\left(\boldsymbol{f}\left(x_b^+\right), y_b^+\right) \tag{15}$$

is the "fix" loss. $(x_b^+, y_b^+)$ represents for a sample $x_b^+$ with high confidence pseudo label $y_b^+$ and

$$\mathcal{L}_{\text{mix}} = \lambda_{\text{mix}} \cdot \ell\left(\boldsymbol{f}\left(x_{\text{mix}}\right), y_b^+\right) + (1 - \lambda_{\text{mix}}) \cdot \ell\left(\boldsymbol{f}\left(x_{\text{mix}}\right), y_b^-\right)) \tag{16}$$

is the "mix" loss, in which

$$\lambda_{\text{mix}} \sim \text{Beta}(a, a), \quad x_{\text{mix}} \leftarrow \lambda_{\text{mix}} x_b^+ + (1 - \lambda_{\text{mix}}) x_b^-. \tag{17}$$

$a$ is the mixup hyper-parameter and $(x_b^-, y_b^-)$ represents for a sample $x_b^-$ with low confidence pseudo label $y_b^-$. Following the default implementation in Diao et al. (2021), we set hyper-parameter $a = 0.75$. Moreover, we set the weight of "mix" loss $\lambda = 0.3$ and the confidence threshold $\tau = 0.4$.

| Layer | Details |
|-------|---------|
| 1 | Conv2D(3, 64, 7, 2, 3), BN(64), ReLU |
| 2 | Conv2D(64, 64, 3, 1, 1), BN(64), ReLU |
| 3 | Conv2D(64, 64, 3, 1, 1), BN(64) |
| 4 | Conv2D(64, 64, 3, 1, 1), BN(64), ReLU |
| 5 | Conv2D(64, 64, 3, 1, 1), BN(64) |
| 6 | Conv2D(64, 128, 3, 2, 1), BN(128), ReLU |
| 7 | Conv2D(128, 128, 3, 1, 1), BN(64) |
| 8 | Conv2D(64, 128, 1, 2, 0), BN(128) |
| 9 | Conv2D(128, 128, 3, 1, 1), BN(128), ReLU |
| 10 | Conv2D(128, 128, 3, 1, 1), BN(64) |
| 11 | Conv2D(128, 256, 3, 2, 1), BN(128), ReLU |
| 12 | Conv2D(256, 256, 3, 1, 1), BN(64) |
| 13 | Conv2D(128, 256, 1, 2, 0), BN(128) |
| 14 | Conv2D(256, 256, 3, 1, 1), BN(128), ReLU |
| 15 | Conv2D(256, 256, 3, 1, 1), BN(64) |
| 16 | Conv2D(256, 512, 3, 2, 1), BN(512), ReLU |
| 17 | Conv2D(512, 512, 3, 1, 1), BN(512) |
| 18 | Conv2D(256, 512, 1, 2, 0), BN(512) |
| 19 | Conv2D(512, 512, 3, 1, 1), BN(512), ReLU |
| 20 | Conv2D(512, 512, 3, 1, 1), BN(512) |
| 21 | AvgPool2D |
| 22 | FC(512, 10) |

Table D.3: Model architecture of the benchmark experiment on CIFAR10. For convolutional layer (Conv2D), we list parameters with sequence of input and output dimension, kernel size, stride and padding. For max pooling layer (MaxPool2D), we list kernel and stride. For fully connected layer (FC), we list input and output dimension. For BatchNormalization layer (BN), we list the channel dimension.

Table D.4: Detailed weight of L1 regularization used in benchmark experiments.

| Dataset | MNIST | | | CIFAR10 | | |
|---------|-------|-------|-------|---------|-------|-------|
| Sets | 10 | 20 | 30 | 10 | 20 | 30 |
| Weight | 1e-5 | 5e-5 | 2e-6 | 2e-5 | 1e-5 | 4e-6 |

- FedLLP (Dulac-Arnold et al., 2019): this baseline method is based on empirical proportion risk minimization. The learning objective is the aggregated proportion risk, i.e.,

$$\widehat{R}(\boldsymbol{f}) = \frac{1}{C} \sum_{c=1}^{C} \sum_{m=1}^{M_c} d_{\text{prop}} \left( \pi_c^m, \widehat{\pi}_c^m \right) \tag{18}$$

where $\pi_c^m$ and $\widehat{\pi}_c^m$ are the true and predicted label proportions for the $m$-th U set $\mathcal{U}_c^m$ at the $c$-th client, $\widehat{\pi}_c^m := \frac{1}{M_c} \sum_{\boldsymbol{x} \sim \mathcal{U}_c^m} f(\boldsymbol{x})$. For each client $c$, $d_{\text{prop}}$ is defined as

$$d_{\text{prop}}(\pi_c^m, \widehat{\pi}_c^m) := - \sum_{k=1}^{K} \pi_c^m \log \widehat{\pi}_c^m.$$

- FedLLP-VAT (Tsai & Lin, 2020): this method is based on empirical proportion risk minimization, integrated with a consistency regularization term. The learning objective is the

Table D.5: Detailed number of samples at each client for MNIST benchmark

|  | Class | 0 | 1 | 2 | 3 | 4 | 5 | 6 | 7 | 8 | 9 |
|---|---|---|---|---|---|---|---|---|---|---|---|
| IID | Client1 | 920 | 1098 | 970 | 972 | 945 | 893 | 898 | 1007 | 971 | 926 |
|  | Client2 | 920 | 1098 | 970 | 972 | 945 | 893 | 898 | 1007 | 971 | 926 |
|  | Client3 | 920 | 1098 | 970 | 972 | 945 | 893 | 898 | 1007 | 971 | 926 |
|  | Client4 | 920 | 1098 | 970 | 972 | 945 | 893 | 898 | 1007 | 971 | 926 |
|  | Client5 | 920 | 1098 | 970 | 972 | 945 | 893 | 898 | 1007 | 971 | 926 |
| Non-IID | Client1 | 4498 | 4646 | 56 | 56 | 54 | 57 | 54 | 60 | 60 | 54 |
|  | Client2 | 57 | 54 | 4425 | 4737 | 49 | 50 | 49 | 58 | 57 | 58 |
|  | Client3 | 64 | 61 | 57 | 61 | 4316 | 4217 | 54 | 53 | 58 | 57 |
|  | Client4 | 53 | 58 | 55 | 35 | 50 | 0 | 4572 | 4576 | 60 | 57 |
|  | Client5 | 57 | 58 | 61 | 0 | 60 | 0 | 19 | 54 | 4418 | 4576 |

Table D.6: Detailed number of samples at each client for CIFAR10 benchmark

|  | Class | airplane | automobile | bird | cat | deer | dog | frog | horse | ship | truck |
|---|---|---|---|---|---|---|---|---|---|---|---|
| IID | Client1 | 755 | 788 | 794 | 796 | 791 | 842 | 827 | 788 | 820 | 799 |
|  | Client2 | 755 | 788 | 794 | 796 | 791 | 842 | 827 | 788 | 820 | 799 |
|  | Client3 | 755 | 788 | 794 | 796 | 791 | 842 | 827 | 788 | 820 | 799 |
|  | Client4 | 755 | 788 | 794 | 796 | 791 | 842 | 827 | 788 | 820 | 799 |
|  | Client5 | 755 | 788 | 794 | 796 | 791 | 842 | 827 | 788 | 820 | 799 |
| Non-IID | Client1 | 3748 | 3872 | 47 | 46 | 45 | 47 | 45 | 50 | 50 | 45 |
|  | Client2 | 48 | 45 | 3688 | 3938 | 41 | 41 | 41 | 48 | 48 | 48 |
|  | Client3 | 53 | 51 | 48 | 0 | 3597 | 3887 | 45 | 53 | 48 | 47 |
|  | Client4 | 44 | 18 | 46 | 0 | 42 | 0 | 3810 | 3813 | 50 | 48 |
|  | Client5 | 47 | 0 | 51 | 0 | 50 | 0 | 46 | 45 | 3788 | 3790 |

aggregated proportion risk with consistency regularization, i.e.,

$$\widehat{R}(\boldsymbol{f}) = \frac{1}{C} \sum_{c=1}^{C} \left( \sum_{m=1}^{M_c} d_{\mathrm{prop}} \left( \pi_c^m, \widehat{\pi}_c^m \right) + \alpha \ell_{\mathrm{cons}} \left( f \right) \right), \tag{19}$$

where $\pi_c^m$ and $\widehat{\pi}_c^m$ are the true and predicted label proportions for the $m$-th U set $\mathcal{U}_c^m$ at the $c$-th client, and $d_{\mathrm{prop}}$ is a distance function.

$$\ell_{\mathrm{cons}} \left( f \right) = d_{\mathrm{cons}} \left( f(\boldsymbol{x}), f(\widehat{\boldsymbol{x}}) \right) \tag{20}$$

is the consistency loss, where $d_{\mathrm{cons}}$ is a distance function, and $\widehat{\boldsymbol{x}}$ is a perturbed input from the original $\boldsymbol{x}$. Following the default implementation in their paper (Tsai & Lin, 2020), we set the hyper-parameter, the weight of consistency loss $\alpha = 0.05$ and the perturbation weight $\mu = 6.0$ for FedLLP-VAT on CIFAR10 benchmark experiment. For MNIST benchmark, we set the hyper-parameter, the weight of consistency loss $\alpha = 5e - 4$ and the perturbation weight $\mu = 1e - 2$.

**Real-world Experiments:** We perform a case study on the clinical task — intracranial hemorrhage diagnosis. We use the RSNA Intracranial Hemorrhage Detection dataset (Flanders et al., 2020) (Figure 4), which contains large-scale Magnetic Response Image (MRI) data from several hospitals.There are $106k$ samples in the dataset belonging to six classes, including five types of intracranial hemorrhage (Epidural, Intraparenchymal, Intraventricular, Subarachnoid, and Subdural) and healthy control type. In our experiments, we simulate a real-world scenario with five healthcare systems (a.k.a. clients) and assume the number of hospitals (a.k.a sets) in each healthcare systems varies $M = 12, 24$, and $48$. At first, we randomly split the whole dataset for five healthcare systems equally, so each healthcare systems is assigned $21.2k$ samples. Then we generate $\pi_c^m$ for $m$-th set in $c$-th client from range $[0.1, 0.9]$ for all the hospitals with U data only. We check that the generated class priors are not all identical. Finally, we load samples for each the hospitals following the generated $\pi$. Detailed dataset statistics is given in Table D.7.

During training process, we use Adam optimizer with learning rate $1e - 4$ and cross-entropy loss. We set batch size to $128$ and training rounds to $100$. To avoid overfitting, we use L1 regularization. We

Table D.7: Detailed number of samples in each healthcare system (HS) for real-world RSNA Intracranial Hemorrhage Detection dataset (Flanders et al., 2020).

| Type | Healthy | Epidural | Intraparenchymal | Intraventricular | Subarachnoid | Subdural |
|---|---|---|---|---|---|---|
| HS 1 | 3726 | 221 | 2023 | 1198 | 2027 | 4005 |
| HS 2 | 3756 | 226 | 1923 | 1190 | 2063 | 4042 |
| HS 3 | 3804 | 177 | 1932 | 1266 | 2040 | 3981 |
| HS 4 | 3805 | 224 | 1939 | 1187 | 2074 | 3971 |
| HS 5 | 3711 | 198 | 1929 | 1299 | 2080 | 3983 |

set the weight of L1 regularization as $2e - 6$, $8e - 7$ and $4e - 7$ for 10, 20 and 40 sets, respectively. In real-world experiments, we use the same hyper-parameters as benchmark experiments for all methods except FedLLP-VAT. For FedLLP-VAT, we set the weight of consistency loss $\alpha = 0.05$ and the perturbation weight $\mu = 6.0$.

**Training Efficiency:** We further evaluate the training efficiency of our method with comparison to baseline methods. The detailed one epoch training time (mean) and forward FLOPs are given in Table D.8. The computation time is evaluated on a PC with one Nvidia TITAN X GPU. We can see that our method slightly increase the forward FLOPs and can achieve higher training efficiency compared with FedPL and FedLLP-VAT.

Table D.8: Quantitative comparison of our method with the baseline methods training epoch time and forward flops on benchmark and real-world datasets under Non-IID and IID setting.

| Dataset | Method | Non-IID Task with 5 Clients | | IID Task with 5 Clients | |
|---|---|---|---|---|---|
| | | Computation time (s) | FLOPs (G) | Computation time (s) | FLOPs (G) |
| MNIST | FedPL | 3.29 | 0.048125312 | 4.78 | 0.048125312 |
| | FedLLP | 1.67 | | 2.45 | |
| | FedLLP-VAT | 2.10 | | 2.87 | |
| | FedUL | 1.80 | 0.048125422 | 2.35 | 0.048125422 |
| CIFAR10 | FedPL | 3.37 | 0.141595648 | 5.70 | 0.141595648 |
| | FedLLP | 1.95 | | 2.44 | |
| | FedLLP-VAT | 3.43 | | 4.86 | |
| | FedUL | 2.63 | 0.141595758 | 3.80 | 0.141595758 |
| Real-world | FedPL | 32.47 | 6.937938944 | - | - |
| | FedUL | 32.34 | 6.937939028 | - | - |

# E   MORE EXPERIMENTAL RESULTS ON THE BENCHMARK DATASETS

## E.1   DETAILED STATISTICS OF FIGURE 3(A)

We show detailed statistics of Figure 3(a) in Table E.9. With different noisy class priors, even with challenging $\epsilon = 0.8$ and $\epsilon = 1.6$, our method still achieves low error rate, which verify the robustness and effectiveness of our method.

Table E.9: Inaccurate class priors experiments with different noise ratio $\epsilon$ on MNIST under Non-IID setting with 5 clients and 10 sets for every client (mean error (std)).

| $\epsilon = 0$ (Clean) | $\epsilon = 0.2$ | $\epsilon = 0.4$ | $\epsilon = 0.8$ | $\epsilon = 1.6$ |
|---|---|---|---|---|
| 2.98 (1.72) | 3.02 (1.30) | 3.10 (0.96) | 3.02 (1.36) | 3.49 (1.74) |

## E.2   DETAILED STATISTICS OF FIGURE 3(B)

We show detailed statistics of Figure 3(b) in Table E.10. With more Non-IID data, our method performs better, which validates that more unlabeled Non-IID data can be helpful for FL.

Table E.10: Experiments on accessible data amount under the Non-IID setting (mean error (std)).

| 5k | 10k | 15k | 20k | 25k |
|---|---|---|---|---|
| 3.56 (0.09) | 2.62 (0.07) | 1.97 (0.38) | 1.75 (0.11) | 1.72 (0.13) |

## E.3   DIFFERENT COMBINATIONS OF $E$ AND $B$

We compare our method with different combinations of $E$ and $B$ on benchmark dataset MNIST in the Non-IID setting. The classification error rate (mean and std) is consistently small as presented (cf. Table E.11), which indicates our method can work under a wide range combinations of $E$ and $B$.

Table E.11: Experiments using different combinations of training batch size $B$ and local update epoch $E$ on benchmark dataset MNIST in the Non-IID setting (mean error (std)).

| E |  | 1 |  |  | 4 |  |  | 16 |  |
|---|---|---|---|---|---|---|---|---|---|
| B | 16 | 128 | $\infty$ | 16 | 128 | $\infty$ | 16 | 128 | $\infty$ |
| Error | 7.99 (9.83) | 2,93 (1.20) | 3.69 (0.88) | 2.54 (1.09) | 4.16 (3.44) | 2.81 (0.37) | 2.92 (1.12) | 3.51 (1.88) | 2.88 (1.06) |

## E.4   BENCHMARK EXPERIMENTS ON CLIENTS WITH DIFFERENT SETS

Here we conduct experiments on benchmark datasets under IID and Non-IID setting with 5 clients and different sets for each client (10, 20, 30, 40, 50). The experimental results are shown in Table E.12, which further indicates the effectiveness of our method even under this challenging setting.

## E.5   BENCHMARK EXPERIMENTS WITH MORE CLIENTS

To study the influence of the number of clients, here we further conduct benchmark experiments on both MNIST and CIFAR10 with 10 clients. The detailed results are shown in Table E.13. The observations are similar to Table 1.

Table E.12: Quantitative comparison of our method with the baseline methods on benchmark datasets under Non-IID and IID setting (mean error (std)) with different sets for each client. The best method (paired t-test at significance level 5%) is highlighted in boldface.

**IID Task with 5 Clients**

| Dataset | FedPL | FedLLP | FedLLP-VAT | FedUL | FedAvg 10% |
|---------|-------|--------|------------|-------|------------|
| MNIST | 3.74 (0.78) | 5.22 (5.38) | 2.24 (0.05) | **1.19 (0.11)** | 2.21 (0.07) |
| CIFAR10 | 45.49 (2.74) | 84.16 (0.71) | 86.01 (2.54) | **31.17 (2.80)** | 42.27 (1.06) |

**Non-IID Task with 5 Clients**

| Dataset | FedPL | FedLLP | FedLLP-VAT | FedUL | FedAvg 10% |
|---------|-------|--------|------------|-------|------------|
| MNIST | 12.03 (2.89) | 4.26 (3.20) | 3.88 (2.89) | **2.31 (1.14)** | 3.42 (0.20) |
| CIFAR10 | 59.06 (3.73) | 62.44 (16.99) | 59.72 (18.73) | **35.46 (1.40)** | 53.13 (0.22) |

Table E.13: Quantitative comparison of our method with the baseline methods on benchmark datasets under Non-IID and IID setting (mean error (std)). The best method (paired t-test at significance level 5%) is highlighted in boldface.

**IID Task with 10 Clients**

| Dataset | Sets | FedPL | FedLLP | FedLLP-VAT | FedUL | FedAvg 10% |
|---------|------|-------|--------|------------|-------|------------|
| MNIST | 10 | 2.19 (0.04) | 10.11 (7.12) | 25.83 (22.64) | **0.74 (0.14)** | |
| | 20 | 2.55 (0.30) | 24.23 (17.52) | 28.40 (23.20) | **1.05 (0.05)** | 1.58 (0.23) |
| | 40 | 3.13 (0.25) | 32.53 (16.93) | 34.46 (15.46) | **1.47 (0.11)** | |
| CIFAR10 | 10 | 40.47 (0.58) | 76.23 (1.47) | 76.25 (1.65) | **20.40 (0.63)** | |
| | 20 | 46.44 (0.71) | 78.10 (3.58) | 76.26 (3.83) | **21.61 (0.23)** | 33.57 (1.03) |
| | 40 | 51.58 (1.18) | 79.10 (2.34) | 80.44 (1.60) | **22.71 (0.17)** | |

**Non-IID Task with 10 Clients**

| Dataset | Sets | FedPL | FedLLP | FedLLP-VAT | FedUL | FedAvg 10% |
|---------|------|-------|--------|------------|-------|------------|
| MNIST | 10 | 12.24 (1.36) | 20.81 (14.72) | 17.26 (10.39) | **1.91 (0.16)** | |
| | 20 | 5.73 (0.89) | 17.67 (11.48) | 18.45 (4.30) | **1.45 (0.27)** | 3.03 (0.24) |
| | 40 | 4.16 (0.56) | 14.98 (10.11) | 13.62 (5.47) | **1.27 (0.12)** | |
| CIFAR10 | 10 | 68.53 (1.14) | 84.11 (2.63) | 82.17 (2.58) | **39.20 (0.79)** | |
| | 20 | 61.49 (1.42) | 71.69 (7.59) | 72.34 (7.39) | **34.12 (0.57)** | 52.48 (1.55) |
| | 40 | 55.89 (2.19) | 58.21 (4.95) | 59.42 (9.59) | **32.21 (1.02)** | |

