# OpenReview forum: "Federated Learning from Only Unlabeled Data with Class-conditional-sharing Clients"
_ICLR.cc/2022/Conference — ICLR 2022 Poster_

### Official Review · Reviewer_mco5 · 2021-10-25

**Correctness:** 4
**Technical Novelty And Significance:** 3
**Empirical Novelty And Significance:** 3
**Recommendation:** 8
**Confidence:** 3

**Main Review:**

Strong:
- relevant topic with clear motivation and useful practical examples
- sufficient experiments
- outlined theoretical derivations + code

Weakness:
- the title is a bit misleading ... because the approach does not cover
  a general framework for unsupervised learning -> maybe the title should be changed

**Summary Of The Paper:**

The paper suggests a new strategy for pseudo-labeling
based on cluster structures in FedLearn. Theoretical
derivations, abl.-studies and experiments are provided
showing the effectivness of the approach.

**Summary Of The Review:**

Unsupervised FedLearn

comments:
- not so much to mention but only some 'concerns'
-  Unsupervised learning is really different to supervised learning.
   In your case you consider local cluster assignments as labels and use
   subsequently a supervised learning scheme to generate predictions.
   This is not really unsupervised because in general SL methods focus
   on margin maximization of the class boundaries ... which is not the same
   as approximating a cluster distribution (so I would say your title is a bit
   wrong). Maybe better: Federated learning with surrogate labeling?
-  Further what happens if you have no uni-modal data as shown in Fig 1?
   In fact your # of labelings may change during the optimization or you
   artificially colaps different distribution into single once if the #of cluster
   is not appropriate
- 'Without any labels, FL becomes significantly harder than before, since it is unclear how to compute
   the local gradients at each client and how to aggregate them for updating the global model  '
   --> well there are unsupervised learning approaches which are gradient based
   ... you only need a differentiable cost function!
-  if the labeling in the experiment (for labeled data) follows the data distribution
   I do not expect a substantial problem ... but if the data are non-unimodal and the
   #of clusters does not fit it may become a challenge to get reliable models
- theory is likely correct and code is provided so reproducibility should be given

---

> ### Author Response · Authors · 2021-11-17
> **Response to Reviewer mco5**
>
> Thank you so much for your time and constructive comments.
> Please find our detailed responses below.

---

> > ### Author Response · Authors · 2021-11-17
> > **If the data are non-unimodal and the #of clusters does not fit it may become a challenge to get reliable models.**
> >
> > To further verify the effectiveness of our method, we added experiments on benchmark datasets under IID and Non-IID settings with 5 clients: the number of classes K=10, but the number of unlabeled datasets M is set differently for each client (we set M to be 10, 20, 30, 40, 50 for the 5 clients respectively). Here are the results (mean errors (std)):
> >
> > **Non-IID Task**
> >
> > | Dataset |     FedPL    |     FedLLP    |   FedLLP-VAT  |       FedUL      |  FedAvg 10%  |
> > |:-------:|:------------:|:-------------:|:-------------:|:----------------:|:------------:|
> > |  MNIST  | 12.03 (2.89) |  4.26 (3.20)  |  3.88 (2.89)  |  **2.31 (1.14)** |  3.42 (0.20) |
> > | CIFAR10 | 59.06 (3.73) | 62.44 (16.99) | 59.72 (18.73) | **35.46 (1.40)** | 53.13 (0.22) |
> >
> > **IID Task**
> >
> > | Dataset |     FedPL    |     FedLLP    |   FedLLP-VAT  |       FedUL      |  FedAvg 10%  |
> > |:-------:|:------------:|:-------------:|:-------------:|:----------------:|:------------:|
> > |   MNIST |  3.74 (0.78) | 5.22 (5.38)  | 2.24 (0.05)  | **1.19 (0.11)**  | 2.21 (0.07)  |
> > | CIFAR10 | 45.49 (2.74) | 84.16 (0.71) | 86.01 (2.54) | **31.17 (2.80)** | 42.27 (1.06) |
> >
> > The experimental results demonstrate the effectiveness of our proposed method (FedUL) even under this challenging setting. We have added the new results and discussions in the revised version (see Appendix F, Table F.12).

---

> > ### Author Response · Authors · 2021-11-17
> > **What happens if you have no uni-modal data as shown in Fig 1? In fact your # of labelings may change during the optimization or you artificially colaps different distribution into single once if the #of cluster is not appropriate.**
> >
> > In our data generation process, the unlabeled data of each set are sampled from a mixture of multiple component distributions as shown in Equation (5), which is in general multi-modal. When the problem is given, the number of components (classes) K and the number of unlabeled sets M at each client are fixed, so there is no need to worry that the number of labelings may change during optimization. Our method uses the indices of unlabeled datasets as surrogate labels (the number of labeling is fixed as M) and then transforms the local model to be compatible with the K-class supervised federated learning loss by Theorem 1. In Figure 1, for illustration purposes, we show an example of M=K, but our method can generally handle all cases where M>=K.

---

> > ### Author Response · Authors · 2021-11-17
> > **In your case, you consider local cluster assignments as labels and use subsequently a supervised learning scheme to generate predictions. This is not really unsupervised because in general SL methods focus on margin maximization of the class boundaries ... which is not the same as approximating a cluster distribution (so I would say your title is a bit wrong). Maybe better: Federated learning with surrogate labeling?**
> >
> > While it is true that our problem setting is not fully unsupervised, we must emphasize that the concept of unsupervised learning is much more general than clustering. Unsupervised learning does not necessarily rely on the cluster assumption and/or approximate the cluster distribution, and models in unsupervised learning may also have some out-of-sample ability rather than only being able to make in-sample prediction. For example, manifold embedding is unsupervised learning which relies on the manifold assumption and approximates the manifold distribution, and principal component analysis and many similar algorithms from signal processing are also unsupervised from the machine learning point of view.
> >
> > In our case, it is sometimes called “unsupervised classification” in the literature, because the underlying learning principle is empirical risk minimization, the model is for predicting the class label but not the cluster membership, learning here is inductive rather than transductive, and most importantly, we do not have any class label for training classifiers which goes even beyond semi-supervised classification so that it belongs more to unsupervised classification. We are working on the federated version of unsupervised classification (besides the unsupervised version of federated classification), where the surrogate labels are the indices of unlabeled datasets but not the local cluster assignments, because we did not use the cluster assumption at all. We have also added a subtitle to reflect that our problem setting is not fully unsupervised.

---

> > ### Author Response · Authors · 2021-11-17
> > **The title is a bit misleading because the approach does not cover a general framework for unsupervised learning, maybe the title should be changed.**
> >
> > Thanks for raising this point! Yes, the proposed approach does not cover a general unsupervised framework, since we rely on the assumption that the unlabeled data distributions share the same class-conditional distributions. To make it clear, we added a subtitle “Unsupervised Federated Learning is Possible: A Case of Class-Conditional-Sharing Clients”. Please see the response to Reviewer SBTk (https://openreview.net/forum?id=WHA8009laxu&noteId=8MuJVK_uux) for a detailed discussion on this assumption.

---

### Official Review · Reviewer_U3W9 · 2021-10-30

**Correctness:** 3
**Technical Novelty And Significance:** 3
**Empirical Novelty And Significance:** 2
**Recommendation:** 6
**Confidence:** 2

**Main Review:**

Overall there are some good theoretical results in the paper with a thoroughly analyzed experiment section. I felt that Section 2 is not really needed in the paper as explaining the context of supervised federated learning is not relevent to the contributions in this paper. This could probably be moved to the appendix and then some of the theoretical derivations could be moved to the main body.

From the experiments, it was not exactly clear to me which class prior was used. It says "we randomly sampled class priors from range [0.1, 0.9] and then regularized them to formulate a valid \Pi_c as discussed in Section 3.1.". But it does not appears to explain what distribution the class prior is. Could this be explained.

I felt that U shouldn't be used a shorthand for unsupervised. But I guess this is up to the authors.

**Summary Of The Paper:**

The authors are proposing an approach for unsupervised Federated Learning. The authors propose to use Emperical risk minimizing for unsupervised learning. The authors assign labels to each of the class and uses a prior for the classes so that each of the client can learn without any labels.

The authors have shown theoretical properties for their proposed solution. Theorem 1 showing that there is a map that that exist to transform the data to the classes, Lemma 2 shows that there is a transition function for each of the clients. The authors have also shown convergences of the algorithm by deriving the upper bound.


**Summary Of The Review:**

The paper has provided some novel contribution in terms of the theoretical properties for unsupervised learning. I felt that the paper could have been better written and more experiments could have been conducted to make their results more convincing. But nevertheless, I do think that the results that the authors have presented are novel enough for a publication at ICLR.

---

> ### Author Response · Authors · 2021-11-17
> **Response to Reviewer U3W9**
>
> Thank you so much for your time and constructive comments.
> Please find our detailed responses below.

---

> > ### Author Response · Authors · 2021-11-17
> > **More experimental results and better writing.**
> >
> > Thanks for the suggestion!
> > We have added experiments on
> >
> > - comparisons on computation time in Appendix D.2, Table D.8 (please see the discussion with reviewer SBTk https://openreview.net/forum?id=WHA8009laxu&noteId=vR6c7vn1OxW)
> >
> > - more clients in Appendix G, Table G.13 (please see the discussion with reviewer 4sTZ https://openreview.net/forum?id=WHA8009laxu&noteId=_lrkgzN9-gQ)
> >
> > - clients with different sets in Appendix F, Table F.12 (please see the discussion with reviewer mco5 https://openreview.net/forum?id=WHA8009laxu&noteId=xX0ISkBrDtL).
> >
> > We also added a subtitle and modified the algorithm to avoid some confusion.

---

> > ### Author Response · Authors · 2021-11-17
> > **Not clear which class prior was used. Did not explain what distribution the class prior is.**
> >
> > Sorry for the confusion. To clarify, class prior is also called class-prior probability written as p(y=k). We use the term "class prior" adapted from noisy label learning literature where the class prior actually refers to the mixing proportion (\pi in Equation (5)) to be known as a type of prior knowledge, which is not to be confused with the prior in the Bayesian sense. In fact, our prior is about data and our optimal model parameters are fixed (but unknown), while in Bayesian the prior is about the model parameters and the data is considered fixed (because repeated sampling is a concept in frequentist).
> >
> > Thanks a lot for pointing this out: "we randomly sampled class priors from the range [0.1, 0.9]...", but it does not appear to explain what distribution the class prior is. It is the uniform distribution. We actually mean to uniformly select a class mixing proportion between 0.1 to 0.9 to avoid degeneracy for a particular experiment setup, and we have many experiments with different mixing proportions. We have modified the unclear writing.

---

> > ### Author Response · Authors · 2021-11-17
> > **Section 2 explaining the context is not relevant to the contributions, could be moved to appendix.**
> >
> > We provide a detailed introduction of supervised federated learning (especially notations) in order to make Section 3 easier to read. Thanks for the suggestion, we have simplified Section 2 and used the room for adding more theoretical discussions.

---

> ### Author Response · Authors · 2021-11-23
> **Need further clarification?**
>
> Thanks very much for your constructive comments on our work!
> As the rebuttal deadline is approaching, is there still any unclear point (e.g., the class prior) about the paper and the rebuttal?
> Although there is no more time for new experiments as a rebuttal revision, do you have suggested experiments for the next version?

---

### Official Review · Reviewer_4sTZ · 2021-11-01

**Correctness:** 4
**Technical Novelty And Significance:** 4
**Empirical Novelty And Significance:** 4
**Recommendation:** 8
**Confidence:** 4

**Main Review:**

# Pros

-The proposed method is novel and interesting. By using the surrogate loss, the method provides a framework to enable unsupervised clients (without any labels) to supervised clients (with labels in particular form); and learning is performed via the surrogate supervised learning loss. With such a framework, many existing supervised federated learning methods can also be applied in this client-without-label setting. I think this framework provides a novel perspective for solving unsupervised federated learning, instead of following the unsupervised learning approach such as clustering.

-It is also nice that the method can be easily implemented by adding a transition layer to the existing models, neither changes on the optimization process nor introduces additional hyperparameters to be tuned.

-The theoretical analysis on the optimal model recovery is also a very useful result providing both conditions for unsupervised surrogate task to match the supervised task optimal; and the interpretations for these surrogates. Its convergence directly relies on the chosen federated aggregation approach, which has quite a flexibility on advanced methods with guarantees.


# Cons

-The discussions in the related work section may not be comprehensive. The work can be quite related to personalized federated learning in the sense that each client learns a personalized model (g1,...,gc), see:
Towards Personalized Federated Learning, Tan et al., 2021.
Discussing relations and differences with this line of research seems necessary.

-Another line of research working on utilizing unlabeled data in federated learning follows the semi-supervised learning framework, see
A Survey towards Federated Semi-supervised Learning, Jin et al., 2020.
Although they assume some labeled data are available, which is not exactly the unsupervised federated learning setting, adding discussions with them in the related work may help position the paper well.

-In this paper, the authors do assume certain conditions on the unlabeled data, e.g. the assumptions on the class prior knowledge on each unlabeled dataset. Can the authors give more examples on how this information can be obtained in the real world?

# Minor comments

-In algorithm 1, it seems that only the model f is trained. In my understanding, equation 7 is the surrogate loss and model g is what is being trained. If it is correct, the current algorithm may be a bit misleading.

-In table 1, the performances of FedPL and FedUL are reasonable when the set number changes. However FedLLP and FEdLLP-VAT are not very stable. Could the authors explain more on this phenomenon?

-In the experiments, it seems that the experiments are conducted only on 5 clients, could the authors show more results with more clients?



**Summary Of The Paper:**

This work presents a novel federated learning scheme to address the problem of learning from only unlabeled data. The main idea is clean and interesting, which constructs a global (server) model by aggregating the surrogate clients’ tasks from observing only unlabeled data for the classification tasks. The unlabeled data are transformed at each client to make them compatible with supervised federated learning; consequently, the learned models are transformed accordingly. Existing federated aggregation techniques, e.g. FedAvg, are applied on these transformed clients (where they call surrogate clients). Theoretical results are provided for learning the optimal model and experiments on benchmark and real data show superior performance compared to baselines.


**Summary Of The Review:**

Overall, this paper is novel and interesting. However, some points should be improved in the revision.

---

> ### Author Response · Authors · 2021-11-17
> **Response to Reviewer  4sTZ**
>
> Thank you so much for your time and constructive comments.
> Please find our detailed responses below.

---

> > ### Author Response · Authors · 2021-11-17
> > **Answers to other minor points.**
> >
> > Q: In algorithm 1, it seems that only the model f is trained. In my understanding, equation 7 is the surrogate loss and model g is what is being trained. If it is correct, the current algorithm may be a bit misleading.
> >
> > A: Sorry for the confusion. Yes, Equation (7) is the loss we use for training, and g is the model being trained. Note that Equation (7) can be reformulated as Equation (9), therefore the update on g naturally induces an update on f. We have modified algorithm 1 to make it clearer.
> >
> > Q: Why FedLLP and FedLLP-VAT are not stable in experiments?
> >
> > A: The LLP methods are based on empirical proportion risk minimization, which aims at predicting accurate class proportions for a set instead of the accurate label for each instance. These methods may suffer from non-identifiability issues in experiments and have no guarantees on classification performances. In contrast, FedPL and FedUL are based on empirical risk minimization which is the common practice for classification and has theoretical guarantees.
> >
> > Q: More experimental results with more clients.
> >
> > A: This has been explored in Figure 3 (b), where the number of clients has been increased from 10 to 50 on MNIST under a Non-IID setting. We further conducted more experiments increasing the number of clients in Table 1. The experimental results are reported in Appendix G, Table G.13 and the observations are similar to Table 1.

---

> > ### Author Response · Authors · 2021-11-17
> > **More examples on how the class prior knowledge can be obtained in the real world.**
> >
> > The class prior knowledge can be obtained from pre-existing data sources, for example, [1] uses census data as class priors for country-level bags of Twitter data; [2] uses recorded proportions of cancer subtypes as class priors. It can also be obtained by asking some domain experts or estimated by labeling a small amount of data, which is much cheaper than obtaining all the ground-truth labels.
> >
> > References
> >
> > [1] Mining the Demographics of Political Sentiment from Twitter Using Learning from Label Proportions, Ardehaly et al., ICDM2017
> >
> > [2] Negative Pseudo Labeling using Class Proportion for Semantic Segmentation in Pathology, Tokunaga et al., ECCV2020

---

> > ### Author Response · Authors · 2021-11-17
> > **Differences with semi-supervised federated learning.**
> >
> > We agree with the reviewer that the motivation of our work can be related to semi-supervised federated learning since both work on utilizing unlabeled data in federated learning. However, our learning settings are different since they assume some labeled data and we do not as the reviewer mentioned, and our methods are different since they are based on semi-supervised regularizations and we are based on empirical risk minimization as reviewer SBTk mentioned (https://openreview.net/forum?id=WHA8009laxu&noteId=a5I4RAe0bZr).

---

> > ### Author Response · Authors · 2021-11-17
> > **Differences with the existing personalized federated learning.**
> >
> > A good question! Indeed, at first glance, our work looks quite similar to personalized federated learning since each client learns a “personalized” model g_c (c is the index for clients), which contains a global model f and client-specific transition layer Q_c. However, our learning settings and methods are very different from the existing personalized approaches.
> >
> > Firstly, the existing personalized federated learning specifically aims to address the challenges of statistics and system heterogeneity. However, our framework focuses on unsupervised federated learning. In our work, the “personalized” model g_c only serves as a proxy, and our ultimate goal is to learn f, which is not exactly personalized. Specifically, 1) If inter-clients are iid, we have Q_1 = ... = Q_c. This yields the same g_c and f for all the clients; 2) If inter-clients are non-iid, although g_c are different (as Q_c are different), the underlying model f (what we want) is still the same.
> >
> > Secondly, existing personalized FL methods adapt the global model for different clients based on user context, knowledge distillation, multi-task learning, meta-learning, etc. Our method differs from them by its surrogate training nature: it works as a wrapper that transforms the unlabeled clients to (surrogate) labeled ones and is compatible with many supervised federated learning methods.
> >
> > We did not include these discussions in the original version due to limited space. Thanks for the reviewer’s suggestion, we are happy to cite related papers and have added more related work in Appendix H.

---

> ### Author Response · Authors · 2021-11-19
> **Need further clarification?**
>
> Thanks very much for your constructive comments on our work. We have tried our best to address the concerns. Is there any unclear point so that we should/could further clarify?

---

> > ### Comment · Reviewer_4sTZ · 2021-11-22
> > **Thanks for the clarification**
> >
> > My concerns have been well addressed.

---

### Official Review · Reviewer_SBTk · 2021-11-01

**Correctness:** 3
**Technical Novelty And Significance:** 3
**Empirical Novelty And Significance:** 3
**Recommendation:** 8
**Confidence:** 4

**Main Review:**

The proposed method allows different clients to have different unlabeled data distributions. There are two conditions required for learning from such data. First, all unlabeled data distributions must share the same set of class-conditional distributions. Second, the ratios of classes must be given, even though the data are unlabeled. Then, with the additional information about the class ratios, each client could align with the supervised learning counterpart itself, and the server would not be affected by the fact that the training data are all unlabeled.

It is worth mentioning that unsupervised federated learning is still based on empirical risk minimization (the loss function is new and the regularization is optional). This means that it goes along different lines from semi-supervised federated learning based on semi-supervised regularizations (the loss function is the same as supervised federated learning and the regularization is mandatory).

Positive points:

1. The idea is novel. The underlying task is still multi-class classification rather than clustering. As far as I know, the problem has not been studied yet, though there is already semi-supervised federated learning.

2. The idea is general. The paper proposed a new loss function, and thus it does not affect our choices of the deep network and optimizer used for training. Moreover, the new loss itself is fine with both image data and text data, which is more general than many semi-supervised techniques that can only be used with image data.

3. The idea is motivated. Since all clients have to prepare training data by themselves and no client would like to share its data with any other party, it is natural that the clients do not want to label all the unlabeled data due to the money and/or time concerns.

Negative points:

1. The assumption that "all unlabeled data distributions must share the same set of class-conditional distributions" sounds strong to me. How can the authors extend the current method so that this assumption is removed or reduced to a milder assumption?

2. The new loss function leads to some additional operations in forward and backward passes. However, in section 4 experiments, there is no experimental result about the computational efficiency. I am working on the practical side of deep learning and I think the issue of computational efficiency is very important. In tables 1 and 2, the proposed method worked much better than the baseline methods, but might the proposed method be much slower than the baseline methods too?

**Summary Of The Paper:**

The paper proposed federated learning in an unsupervised manner with some nice theoretical guarantees and nice experimental results, where the unlabeled data distributions must satisfy some non-trivial assumptions.

**Summary Of The Review:**

This is an overall well-executed paper. Besides the above comments, the authors need clarify on a few raised questions.

---

> ### Author Response · Authors · 2021-11-17
> **Response to Reviewer SBTk**
>
> Thank you so much for your time and constructive comments. Please find our detailed responses below.

---

> > ### Author Response · Authors · 2021-11-17
> > **Might the proposed method be much slower than the baseline methods? No experimental results about the computational efficiency.**
> >
> > The proposed loss function only requires adding a simple transition layer, which is not trained and determined by the known class priors. To analyze its cost, we computed the forward FLOPs (G) for our method (with transition layer) and baselines (w/o the transition layer) in the benchmark and real-world experiments:
> >
> > |   Dataset  |  Baselines  |  Our method |
> > |:----------:|:-----------:|:-----------:|
> > | MNIST      | 0.048125312 | 0.048125422 |
> > | CIFAR10    | 0.141595648 | 0.141595758 |
> > | Real-world | 6.937938944 | 6.937939028 |
> >
> > We can see FLOPs of this additional optimization **can be ignored** by using GPU for training.
> >
> > We also ran more experiments and compared the computation time (s) of the proposed method with baselines on benchmarks and real-world data:
> >
> > **Non-IID Task**
> >
> > |   Method   | MNIST | CIFAR10 | Real-world |
> > |:----------:|:-----:|:-------:|:----------:|
> > | FedPL      |  3.29 |    3.37 |      32.47 |
> > | FedLLP     |  1.67 |    1.95 |          - |
> > | FedLLP-VAT |  2.10 |    3.43 |          - |
> > | Our method |  1.80 |    2.63 |      32.34 |
> >
> > **IID Task**
> >
> > |   Method   | MNIST | CIFAR10 |
> > |:----------:|:-----:|:-------:|
> > | FedPL      |  4.78 |    5.70 |
> > | FedLLP     |  2.45 |    2.44 |
> > | FedLLP-VAT |  2.87 |    4.86 |
> > | Our method |  2.35 |    3.80 |
> >
> > The results show that the proposed method is not slower than the baselines. In general, the proposed method based on empirical risk minimization (ERM) is as fast as FedLLP based on empirical proportion risk minimization (EPRM). Both of them are faster than FedPL and FedLLP-VAT, which are based on ERM and EPRM integrated with a mixup regularization and a consistency regularization, respectively.
> >
> > We have added the new experimental results and discussions in the revised version (see Appendix D.2, Table D.8).

---

> > ### Author Response · Authors · 2021-11-17
> > **Assumption of sharing class-conditional distributions sounds strong. How can the current method be extended so that the assumption is removed?**
> >
> > In our problem setting, each client has access to multiple sets of unlabeled data, and these sets can be very **different** by changing their class priors. Without any assumptions, it would be impossible to infer the distribution on the test set from the (biased) distributions on the unlabeled training sets [1]. Therefore we assume these unlabeled data distributions share the same class-conditional distributions.
> >
> > It is worth noting that, such a class-conditional-sharing assumption has been **widely used** in the existing literature of label-proportion learning and label-noise learning [1,2,3,4,5], which is often referred to as the mutually contaminated distributions (MCD) model (samples are observed from the corrupted class-conditional distributions).
> >
> > Note also that a special case of MCD where the class-posterior probabilities are corrupted has been a very popular topic in recent years, namely, the class-conditional noise, which has a similar but more strict class-posterior-sharing assumption. Therefore, our class-conditional-sharing assumption should not be too strong within this research area.
> >
> > We thank the reviewer for raising this point and have added a subtitle “Unsupervised Federated Learning is Possible: A Case of Class-Conditional-Sharing Clients” to reflect the assumption. However, extending the current method to mitigate this assumption is beyond the scope of the current paper. It is still an open question how to bridge the unsupervised local model with our wanted supervised federated learning model in this case. We agree that this is a promising direction to go (while in our opinion removing it may still be impossible though) and will try our best to investigate it in the future!
> >
> > References
> >
> > [1] Estimating Labels from Label Proportions, Quadrianto et al., JMLR 2009
> >
> > [2] Classification with Asymmetric Label Noise: Consistency and Maximal Denoising, Scott et al., JMLR 2013
> >
> > [3] Clustering unclustered data: Unsupervised binary labeling of two datasets having different class balances, du Plessis et al., TAAI 2013 (best paper)
> >
> > [4] Learning from Corrupted Binary Labels via Class-Probability Estimation, Menon et al., ICML 2015
> >
> > [5] On the minimal supervision for training any binary classifier from only unlabeled data, Lu et al., ICLR 2019

---

### Author Response · Authors · 2021-11-17
**Revision uploaded**

We would like to thank all reviewers for their helpful comments! We have now updated our submission accordingly.
The key modifications of the revised version are as follows.

Added experiments:

* comparisons on computation time (in Appendix D.2, Table D.8)

* experiments on more clients (in Appendix G, Table G.13)

* experiments on clients with different set numbers (in Appendix F, Table F.12)

Added related work:

* related work on personalized FL and illustrations of the differences from our work (in Appendix H)

Modified writing:

* added a subtitle

* simplified Section 2

* modified Algorithm 1

---

### Decision · Program_Chairs · 2022-01-20

**Decision:**

Accept (Poster)

**Comment:**

The paper demonstrates a case of federated learning with unlabelled but systematically partitioned data between clients. A title along terms like "FL with unlabelled data" would be much better - the considered setting here is not fully unsupervised but relies on the key assumption that while not the labels, at lease the precise label frequencies have to be known on each client, which is a strong assumption (also iid up to the class shift). Semi-supervised FL approaches should also be discusses.

Overall, reviewers all agreed that the paper is interesting, well-motivated and deserves acceptance.
We hope the authors will incorporate the open points as mentioned by the reviewers.